# Learning Temporal Rules from Noisy Time-series Data

## Abstract

Events across a timeline are a common data representation, seen in different temporal modalities. Individual atomic events can occur in a certain temporal ordering to compose higher level composite events. Examples of a composite event are a patient's medical symptom or a baseball player hitting a home run, caused distinct temporal orderings of patient vitals and player movements respectively. Such salient composite events are provided as labels in temporal datasets and most works optimize models to predict these composite event labels directly. We focus on uncovering the underlying atomic events and their relations that lead to the composite events within a noisy temporal data setting. We propose Neural Temporal Logic Programming (Neural TLP) which first learns implicit temporal relations between atomic events and then lifts logic rules for composite events, given only the composite events labels for supervision. This is done through efficiently searching through the combinatorial space of all temporal logic rules in an end-to-end differentiable manner. We evaluate our method on video and healthcare datasets where it outperforms the baseline methods for rule discovery.

## 1 Introduction

Complex time series data is present across many data modalities such as sensors, records, audio, and video data. Typically there are *composite events* of interest in these time series which are composed of other *atomic events* in a certain order (Liu et al., 1999; Chakravarthy et al., 1994; Hinze, 2003). An example is a health symptom that can be observed in a doctor's report. Atomic events, such as patient vitals and medications, and their *temporal relations* dictate an underlying causal rule leading to the composite event symptom. These rules may be unknown but useful to recover (Kovačević et al., 2013; Guillame-Bert et al., 2017).

Recent methods leverage the advances in highly parameterized deep architectures to learn latent representations of atomic event data (Pham et al., 2017; Chen et al., 2018; Choi et al., 2019), with the increasing availability of large temporal datasets. Methods, such as LSTM (Hochreiter & Schmidhuber, 1997) or Transformer (Vaswani et al., 2017) based architectures, provide state-of-the-art performance in terms of composite event inference. However, it is uncertain whether the latent representations learn the underlying causal sequence of events or overfit spurious signals in the training data. Having representations faithful to causal mechanisms is advantageous for interpretability, out-of-distribution generalization, and adapting to smaller data sets. Therefore it is important to leverage parametric models that can handle data noise while providing a mechanism to extract explicit temporal rules (Carletti et al., 2019).

Extracting explicit logic rules has been studied through Inductive Logic Programming (ILP) methods (Muggleton, 1991; Muggleton & De Raedt, 1994) and have been leveraged in parametric fashions as well (Yang et al., 2017; Evans & Grefenstette, 2018; Rocktäschel & Riedel, 2017). ILP starts with set of background knowledge, consisting of *grounded* atoms (i.e. facts which do not contain variables) such as `location(Braves, Atlanta)`, where the predicate `location` determines the relationship between the items `Braves` and `Atlanta`. There are set of labels from which rules should be learned. The task is to construct a set of rules, when executed over the background knowledge, entail the provided labels. Given the label `InLeague(Braves, NL East)` and the background knowledge (Figure 1 ILP Input) as input, a candidate rule is `InLeague(Team, League) := Location(Team, City) ∧`

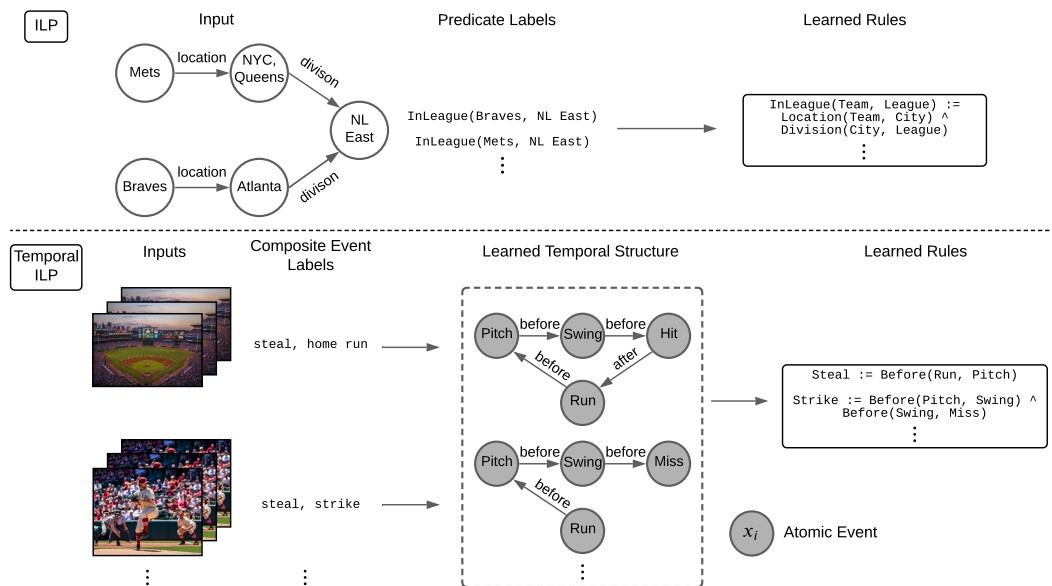

Figure 1: In ILP, the grounded background knowledge and known relational predicates between atoms are provided to induce consistent rules. In the temporal case, we are operating over raw temporal data samples, such as videos, with potentially multiple labels. Therefore the latent temporal structure between the atomic events is recovered, and then the rules for composite events are learned.

`Division(City, League)`. Here `InLeague(Team, League)` is the head of the rule consisting of an atom with variables `Team, League` as items. The body consists of two atoms and when these atoms exist on the background knowledge the rule is evaluated as true. We apply ILP over real world temporal data, however learning such rules poses three key challenges.

**Temporal Background Knowledge**   First, ILP methods operate over an existing grounded background knowledge. The temporal case does not have this knowledge when operating over raw time series. For example in a baseball video, grounded atomic events `pitch` or `swing`, or grounded predicates such as `before(pitch, swing)` are not explicitly provided. By nature, the video would be labeled with a higher level composite event description, such as "Player A's `home run`" instead of individual atomic events and their corresponding temporal predicates. Such atoms can be extracted using a model in a probabilistic fashion at each time point, and a temporal ILP method should handle this uncertainty. The temporal predicates between these probabilistic atomic events can be applied in a rule-based manner (ex. $t_1 < t_2 \rightarrow$ `before`), but due to the noisy nature of extracted atomic events, the predicate predictions should be robust to consistent noise in the atomic event data.

**Atomic Event Relevance**   Second, ILP works learn consistent rules that satisfy a path in the background knowledge given the terms in the labels, such as `InLeague(Braves, NL East)`. The labels are nullary predicates in the temporal case, so the relevant source and target atomic events and predicates to use for rule induction are unknown. In our example, we know from the video we have a label `strike`, but are not told when it occurred or what other events, such as `pitch`, `swing`, and `miss` are needed to compose a rule for `strike`.

Without a prior on which atomic events to search from, we must consider all pairwise temporal relations between atomic events in the input. This leads to a combinatorial search of all pairwise events for *each* predicate in the temporal rule body.

**Multi-Event Labels**   Third, ILP domains work on disjoint labels, while in time series, multiple composite events could occur in each input. In our baseball video, such as a highlight reel, composite event labels `strike`, `steal` and their corresponding atomic events can co-occur in a single video. This further extends the search space of atomic events we consider for each composite event rule.

We illustrate these differences in Figure 1 and further discuss these challenges regarding search complexity in Appendix A. To address these challenges, Neural TLP operates on two key steps.

**Parameter Learning**   First Neural TLP inputs probabilistic atomic events and learns parameters to infer temporal predicates between atomic events. We represent the atomic event data in an interval-based representation to efficiently predict all pairwise predicates between atomic events. The inferred predicates are then projected to predict the composite event labels.

**Structure Learning**   When the predicate parameters are learned, Neural TLP learns a sparse vector to select the correct rule over the combinatorial space of possible rules. To prune the search space, we use the learned projected weights to select candidate grounded predicates per composite event.

We evaluate our method on a synthetic video dataset to empirically test our temporal rule induction performance. Additionally, we apply our framework to provide relevant rules in the healthcare domain, which were verified by doctors.

## 2 PROBLEM FORMULATION

We define the complete set of atomic events $\mathcal{X} = \{x_1, x_2, \ldots, x_{|\mathcal{X}|}\}$ along a timeline $\mathcal{T}$. These atomic events can be existing features in time series data or user defined features of interest. A temporal logic rule $r(\mathcal{X}_r, \mathcal{T}_r)$ can be defined as using a subset of $N \leq |\mathcal{X}|$ atomic events $\mathcal{X}_r = \{x_u\}_{u=1}^N \subseteq \mathcal{X}$, and their associated time intervals $\mathcal{T}_r = \{t_u\}_{u=1}^N \subseteq \mathcal{T}$. The time intervals consists of start and end times $t_u = [t_{u_{\text{start}}}, t_{u_{\text{end}}}]$.

These intervals indicate *durational events* and we can also initialize *instantaneous events* occurring at one time point where $t_{u_{\text{start}}} = t_{u_{\text{end}}}$. A rule is evaluated as true if the corresponding atomic events $x_u$ are present and are in correct ordering with respect to the intervals $t_v$ of other events $x_v$:

$$r(\mathcal{X}_r, \mathcal{T}_r) := ( \bigwedge_{x_u \in \mathcal{X}_r} x_u ) \bigwedge ( \bigwedge_{t_u, t_v \in \mathcal{T}_r} p_i(t_u, t_v))$$

The temporal predicates $p_i \in \{\texttt{before, during, after}\} = \mathcal{P}$ represent a simplified subset of Allen's Temporal Algebra (Allen, 1983). We simplify the notation of the rules as a conjunction of temporal predicates between observed events, where the event time intervals are implicit:

$$r := \bigwedge_{x_u, x_v \in \mathcal{X}_r}^{n} p_i(x_u, x_v) \tag{1}$$

For example, the grounded predicate $\texttt{before}(\texttt{pitch}_{[2,2.7]}, \texttt{swing}_{[3,3.5]})$ would evaluate to true.

These underlying causal rules $r$ induce the composite event labels $r \rightarrow y_r$ seen in the data. Multiple composite events of interest can co-occur during the same time series sample $\mathcal{T}$ which we denote as $\mathbf{y} = \{y_r\}^{|\mathcal{R}|} \in \{0,1\}^{|\mathcal{R}|}$. Any $y_r = 1$ indicates the latent rule $r$ occurred over $\mathcal{T}$ resulting in label $y_r$.

While $\mathcal{T}$ contains precise atomic event interval information, the *observed* time series $\tilde{\mathcal{T}}$ consists of a sequence of probabilistic atomic events from times $[1, T]$. Potentially $k$ different objects $\tilde{\mathcal{T}}^i$ compose the final time series data $\tilde{\mathcal{T}} = \bigcup_{i=1}^k \tilde{\mathcal{T}}^i$. Examples of objects can be multiple concurrent sensor data, or tracking multiple people moving within a video. Then the input $\tilde{\mathcal{T}}$ is formulated as $\mathbf{M_T} \in [0, 1]^{k \times |\mathcal{X}| \times T}$ across object, atomic event, and probability dimensions respectively.

The temporal ILP task is to recover all underlying rules $\mathcal{R}$ given $m$ samples of inputs and labels $\{(\mathbf{M}_{\mathbf{T}i}, \mathbf{y}_i)\}_{i=1}^m$. In Neural TLP this involves learning parameters for the predicates between atomic events and then learning the combination of grounded predicates that induce each $r \in \mathcal{R}$.

## 3 NEURAL TEMPORAL LOGIC PROGRAMMING

Neural TLP operates in two stages. The parameter learning stage learns how to compress the temporal data and learns parameterized temporal predicates. Once these parameters are learned, the structure

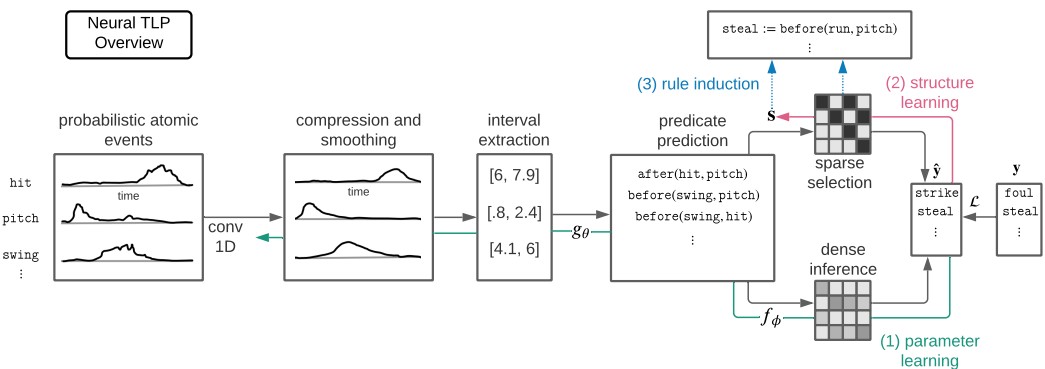

Figure 2: The first step (1) of Neural TLP involves learning the convolution and predicate model parameters from the raw time series and labels. Then the structure learning step (2) is learning attention **s** over a sparse combinatorial matrix to infer the labels. This attention and sparse matrix is then used to carry out the final rule induction (3).

learning stage learns which conjunctive combination of pairwise atomic event predicates is associated with each composite event label. This conjunction composes the rule $r$ for label $y_r$ and is jointly computed for all $\mathcal{R}$. An overview of the framework is presented in Figure 2.

## 3.1 PARAMETER LEARNING STAGE

**Temporal Compression**    Starting from the raw probabilistic atomic event data, we first compress the timeline through convolution. This 1D convolution over the temporal dimension compresses and smooths the timeline to mitigate noise from spurious events. Here the convolution kernel $\mathbf{K}^{|\mathcal{X}| \times l}$ of length $l$ is learned per atomic event. We also parameterize $\alpha$ as an extra degree of freedom to scale these convolved scores, which is useful when computing the intermediate predicates downstream.

$$\mathbf{M_C} \in \mathbb{R}^{k \times |\mathcal{X}| \times t} = \alpha \cdot \text{conv\_1D}(\mathbf{M_T} \in [0, 1]^{k \times |\mathcal{X}| \times T}, \mathbf{K}) \tag{2}$$

The time information is incorporated by multiplying the time dimension $\mathbf{M_D}$ into compressed events: $\mathbf{M_A} \in \mathbb{R}^{k \times |\mathcal{X}| \times t} = \mathbf{M_C} \odot \mathbf{M_D}$. Here $\mathbf{M_D}$ has the same dimensions as $\mathbf{M_C}$, but the temporal dimension is enumerated from $[1, t]$, where $\mathbf{M_D}_{:,:,l} = l$. This can be thought as a positional encoding.

For example if we look at the sample compressed scores for a single object $i$ and atomic event $j$ $\mathbf{M_C}_{i,j,6:10} = [.01, .05, .7, .7, .03]$ and $\mathbf{M_D}_{i,j,6:10} = [6, 7, 8, 9, 10]$ then $\mathbf{M_A}_{i,j,6:10} = [.06, .35, 5.6, 6.3, .3]$. Intuitively we can see that from $\mathbf{M_A}_{i,j,6:10}$ that (1) atomic event $j$ occurs when the scores are high at 5.6 and 6.3 and that (2) score 6.3 occurs after score 5.6 due to the multiplied time index. This temporal representation provides a path to compute precise time intervals of atomic event occurrences and define predicates to compare atomic event intervals.

**Predicate Modeling**    From the compressed timelines, we determine the temporal predicates between atomic events. These relations are computed in a pairwise manner for all atomic events $\forall x_u, x_v \in \mathcal{X}$ occurring in object $i$ through a small network which we call Temporal Predicate Network (TPN). For notation sake here, we represent the atomic event $u$'s timeline for object $i$ as $\mathbf{t}_u^i = \mathbf{M_A}_{i,u,:} \in \mathbb{R}^t$ and correspondingly for atomic event $v$. We denote TPN as $g_\theta(\mathbf{t}_u^i, \mathbf{t}_v^i)$, which takes pairwise atomic event timelines and predicts a temporal predicate $p \in \mathcal{P}$ to indicate the relationship between the atomic events.

Methods such as Temporal Relation Networks (Zhou et al., 2018) learn these predicates between video events by sampling frames throughout the video. The timelines can be long in our setting, and events can occur sparsely, making sampling timelines expensive and noisy. To efficiently compute these relations, we would like to recover each event's underlying start and end time intervals. From intervals, we can encode strong inductive biases to predict the predicates. We are working with continuous time series scores in $\mathbf{t}_u^i, \mathbf{t}_v^i$, so the intervals have to be extracted as the first step in TPN.

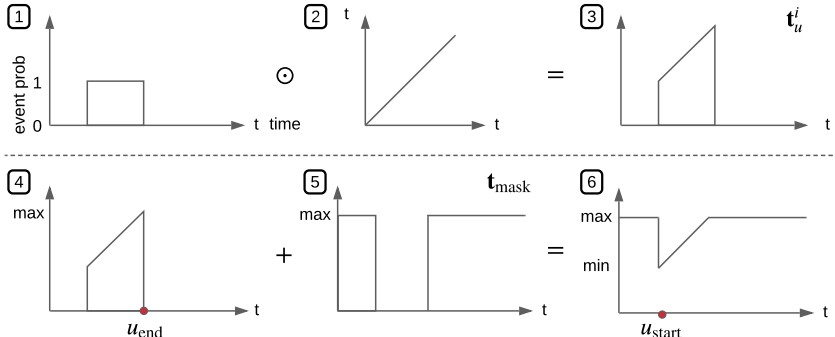

Figure 3: An overview of how intervals are computed from raw data. First the compressed atomic event scores (1) are multiplied with the time scalar (2) to compute $\mathbf{M_A}$ (3), where we observe a single sample vector $\mathbf{t}_u^i$. In step 4 we find the max value of $\mathbf{t}_u^i$, representing the end of the interval, and use this value to initialize the mask in step 5 (Equation 3). When steps 4 and 5 are summed in step 6, we get a representation whose min corresponds to the start of the interval as shown in Equation 4.

To compute the start of an event interval, we create a mask to identify the atomic event noise. Those values will be below some small value $\epsilon$, corresponding to noise in the timeline. We learn the convolution scalar $\alpha$ from Equation 2 to scale scores corresponding to *active* atomic event occurrences above $\epsilon$ while keeping scores corresponding to atomic event *noise* below $\epsilon$. Then the mask is added to the time series, and a min is performed to get the start of the active atomic event interval. Afterwards the min of the mask is subtracted to remove any effect of the mask on the start value.

$$\mathbf{t}_{mask} = (\max(\mathbf{t}_u^i) + \epsilon) \cdot (\mathbf{t}_u^i < \epsilon) \tag{3}$$

$$u_{\text{start}} = \min(\mathbf{t}_u^i + \mathbf{t}_{mask}) - \min(\mathbf{t}_{mask}) \tag{4}$$

To get the end of the event interval we simply compute $u_{\text{end}} = \max(\mathbf{t}_u^i)$ since we multiplied the event scores with the time index earlier. This interval computation from the input time series is visualized in Figure 3. This is computed similarly for the other pairwise event $v$: $[v_{\text{start}}, v_{\text{end}}]$. Given the start and end times for the event pairs $u, v$, the un-normalized predicate scores are computed as:

$$\text{before(u, v)} = v_{\text{start}} - u_{\text{end}} \tag{5}$$

$$\text{after(u, v)} = u_{\text{start}} - v_{\text{end}} \tag{6}$$

$$\text{during(u, v)} = \min(\{v_{\text{end}} - u_{\text{start}}, u_{\text{end}} - v_{\text{start}}\}) \tag{7}$$

Although we use 3 predicates in our model, similar scores can be developed for more fine grained predicates. Then the values are aggregated as $\mathbf{p} = [\text{before(u, v)}; \text{during(u, v)}; \text{after(u, v)}]$ to compute normalized predictions as $\mathbf{p} = \text{softmax}(\frac{\mathbf{p} - \beta}{\gamma})$. Here $\beta$ and $\gamma$ and scale and shift parameters learned from data. Our predicates scores assume that intervals for both $u$ and $v$ occur, so if either event doesn't occur we suppress all predicate predictions:

$$\text{supp} = \min(\{u_{\text{end}} - u_{\text{start}}, v_{\text{end}} - v_{\text{start}}\}) \tag{8}$$

$$\mathbf{p}_i = \min(\{\mathbf{p}_i, \text{supp}\}) \tag{9}$$

Since we leverage a simple interval representation to compare atomic event objects, we can scale comparing atomic events within the object and between the other $k-1$ objects: $x_u \in \mathcal{X}, x_v \in (\mathcal{X} \times k)$. This second-order interaction information is useful if we want to know if, for example, two events occurred simultaneously within different objects. For a single object $i$, these relations are computed for all pairwise predicates through TPN in $\mathbf{M_P} \in \mathbb{R}^{|\mathcal{X}| \times (|\mathcal{X}| \times k) \times |\mathcal{P}|} = \mathbb{R}^{k \times |\mathcal{X}| \times |\mathcal{X}| \times |\mathcal{P}|}$. Aggregating over all objects $k$, we get $\mathbf{M_Q} = [\mathbf{M_{P1}}; \ldots; \mathbf{M_{Pk}}] \in \mathbb{R}^{k^2 \times |\mathcal{X}| \times |\mathcal{X}| \times |\mathcal{P}|}$. We marginalize over the object dimension to get our final pairwise relation matrix $\mathbf{M_R} = \sum_i \mathbf{M_Q}_{i,:,:,:} \in \mathbb{R}^{|\mathcal{X}| \times |\mathcal{X}| \times |\mathcal{P}|}$.

**Composite Event Prediction** The final inference step from the pairwise relational predicates to the composite events labels is carried out by $f_\phi$. This is a linear projection function $f_\phi(\mathbf{M_R}) := \sigma(\text{dropout}(\text{vec}(\mathbf{M_R}))\mathbf{W})$ used to infer the composite event labels $\hat{\mathbf{y}}$.

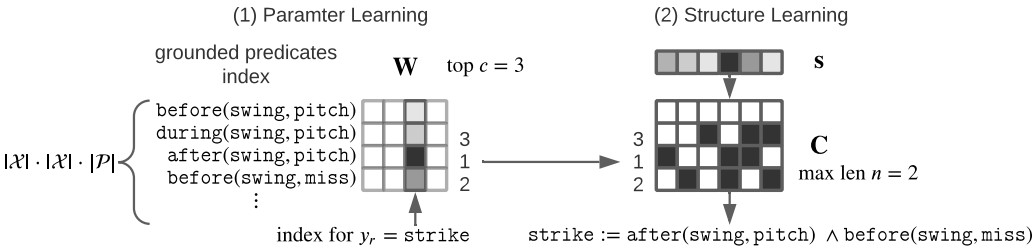

Figure 4: In the parameter learning stage (1) we learn the most relevant grounded predicates used for predicting each label $y_r$ through $\mathbf{W}$. In the structure learning stage (2) we select the most relevant predicates and construct the combinatorial matrix $\mathbf{C}$, where each column indicates a conjunction of predicates. Vector $\mathbf{s}$ is learned to select the most likely conjunction to induce the rule $r$.

Here we flatten $\mathbf{M_R}$ as $\mathrm{vec}(\mathbf{M_R}) \in \mathbb{R}^{|\mathcal{X}| \cdot |\mathcal{X}| \cdot |\mathcal{P}|}$ and regularize it by randomly masking out the grounded predicates (Srivastava et al., 2014). This representation is then projected to the label space using $\mathbf{W} \in \mathbb{R}^{(|\mathcal{X}| \cdot |\mathcal{X}| \cdot |\mathcal{P}|) \times |\mathbf{y}|}$ before passing the un-normalized results through a sigmoid function $\sigma$. $\mathbf{W}$ learns what grounded relational predicates $p_i(x_u, x_v)$, such as `before(pitch, swing)`, correspond to each composite event label. These weights will also be useful for extracting the rules, in the structure learning stage.

## 3.2 STRUCTURE LEARNING STAGE

**Predicate Selection**   To induce the logic rules $\mathcal{R}$, one method is to look at our projection weights $\mathbf{W}$. Here we can select the highest weighted entries corresponding to grounded predicates for each label $y_r$. One can use a conjunction of these predicates to construct the rule $r$ for $y_r$. However, in most cases, the number of predicates needed to compose a rule is unknown apriori. Additionally, setting thresholds for information gain splits is heuristic-based and error-prone, especially when one cannot observe the underlying rules for verification.

**Combinatorial Inference**   Instead of setting thresholds, we directly optimize over the space of combinatorial rules to infer our composite event label $y_r$.

Starting from the predicted grounded predicates $\mathrm{vec}(\mathbf{M_R}) \in \mathbb{R}^d$ where $d = |\mathcal{X}| \cdot |\mathcal{X}| \cdot |\mathcal{P}|$ we initialize a combinatorial matrix up to a max rule body length $n$:

$$\mathbf{C} = [\mathbf{C}_1; ...; \mathbf{C}_n] \; ; \; \mathbf{C}_i \in \mathbb{R}^{d \times \binom{d}{i}} \tag{10}$$

Here for each unique column in $\mathbf{C}_i$ will have indicators for the $i$ chosen predicates, corresponding to one possible combination. Since $\binom{d}{i}$ can be quite large, we sample the top $c < d$ predicate weights in $\mathbf{W}_{:,y_r}$. Then combinations can be initialized over those $c$ predicate indices $\mathbf{C}_i \in \mathbb{R}^{d \times \binom{c}{i}}$.

Selecting the most relevant combination across all combinations is done through an attention vector $\mathbf{a} = \mathrm{softmax}(\mathbf{s})$ where $\mathbf{s} \in \mathbb{R}^{\sum_{i=1}^{n} \binom{c}{i}}$. It is used to weight the combinations $\mathbf{c} \in \mathbb{R}^d = \sum_j a_j \mathbf{C}_{:,j}$ and the label can be inferred by $\hat{y_r} = \mathbf{c}^\top \mathrm{vec}(\mathbf{M_R})$, thus $\hat{\mathbf{y}} = [\hat{y_r}$ for each $r \in \mathcal{R}]$. Note that we maintain *separate* attention parameters $\mathbf{s}$ and unique $\mathbf{C}$ for each label $y_r \in \mathbf{y}$, since each rule relies on different predicates.

**Rule Induction**   To extract the rule we simply choose the column in $\mathbf{C}_j$ where $j = \arg\max \mathbf{s}$ corresponds to the maximum attention value. Each indicator value $i$ in $\mathbf{C}_j^i$ correspond to a grounded temporal predicate $p^i \in \mathcal{P}$ between two events $x_u^i, x_v^i \in \mathbf{X}$, which are used to construct a rule:

$$r := \bigwedge_{i=1}^{n} p^i(x_u^i, x_v^i) \quad \text{for} \quad p^i, x_u^i, x_v^i \in \mathrm{predicate}(\mathbf{C}_j^i) \tag{11}$$

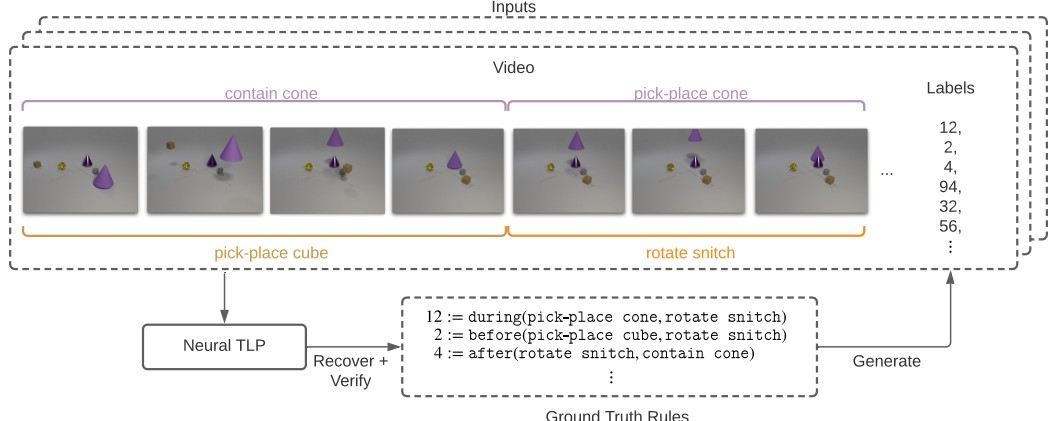

Figure 5: In CATER, generative rules are used to synthesize the labels from the videos. Neural TLP can then learn these rules from the raw atomic event data and labels. We then verify our rule induction performance over the ground truth rules.

This is followed for every composite event label $y_r$ to provide our final set of rules $\mathcal{R}$. This full process is illustrated in Figure 4.

### 3.3 OPTIMIZATION

Now that we have defined our inference procedure to obtain $\hat{\mathbf{y}}$ from both parameter and structure learning stages, we describe the overall training. To train over the data $\{(\mathbf{M_T}, \mathbf{y})\}_{i=1}^{m}$, each stage minimizes the standard cross entropy loss between $\mathbf{y}$ and $\hat{\mathbf{y}}$, denoted as $\mathcal{L}_{ent}$, using the Adam optimizer (Kingma & Ba, 2014) (details in Appendix B).

The parameter learning stage is trained first over the convolution $\alpha, \mathbf{K}$, predicate $\beta, \gamma$, and projection $\mathbf{W}$ parameters. For $\mathbf{W}$ we add $L_1$ regularization (denoted as $\mathcal{L}_1$) and project the weights between $[0, 1]$ to mimic logic weights (Chorowski & Zurada, 2014; Riegel et al., 2020). We also constrain the convolution scalar $\alpha$ between $[0, 1]$. This gives us our final stage objective to optimize $\mathcal{L} = \mathcal{L}_{ent} + \lambda_1 \mathcal{L}_1$ where $\lambda_1 = 0.1$. The structure learning stage is trained next to optimize $\mathcal{L}_{ent}$ over all attention vectors $\mathbf{s}$ corresponding to each label $y_r$, while freezing all other parameters. Each stage is trained in an end-to-end differential manner, after which the temporal logic rules $\mathcal{R}$ are induced.

## 4 EXPERIMENTS

### 4.1 CATER

We explore composite event prediction over complex videos in the CATER dataset (Girdhar & Ramanan, 2019). CATER consists of videos containing objects moving around a scene. The object movements correspond to $|\mathcal{X}| = 14$ distinct atomic events. Every combination of predicates between atomic events yields a rule of length $n = 1$, and when this combination occurs in the video, it induces that corresponding label. There are $|\mathcal{R}| = 301$ rules to recover and the average number of labels per video $\bar{\mathbf{y}} = \frac{1}{m} \sum_{i=1}^{m} \sum_{y_r \in \mathbf{y}_i} y_r = 53$ out of 301.

In previous works on the CATER, the main metric is mean average precision (mAP) over the labels. Due to its synthetic nature, we know the *ground truth* rules used to induce the labels, so we can also empirically evaluate how well our models can recover these rules in its top k rule predictions (Hits@k). The overall task for CATER is illustrated in Figure 5.

**Baselines** We test Neural TLP against two baselines. For the first baseline, we input our $\mathbf{M_T}$ matrix into attention-based LSTM (Hochreiter & Schmidhuber, 1997) and predict the composite events. Since we are dealing with only single predicate rules, we synthesize each rule combination within $\mathbf{M_T}$ and assign it to the highest weighted label.

We define the second baseline as Temporal MAP, which uses the same Neural TLP model. We freeze the parameters and the weight $\mathbf{W}$ is a count of co-occurring grounded relational predicates and labels. This setup is akin to processing atomic event relations *deterministically* and computing $\mathbf{W}$ through MLE. The rule extraction follows the same methods as Neural TLP, and additional baseline details are laid out in Appendix C.

### 4.1.1 RESULTS

| Model | Hits@1 | Hits@5 |
|---|---|---|
| LSTM | .00 | .04 |
| LSTM Attn | .00 | .04 |
| Temporal MAP | .27 | .28 |
| Neural TLP | **.91** | **.95** |

Table 1: Hits when the inputs are inferred (probabilistic) the atomic events. All scores have a reported variance of $\leq .01$.

| Model | Inputs | mAP |
|---|---|---|
| I3D/R3D | ResNet Features + Optical Flow | .44 |
| TSN | RGB Difference + Optical Flow | .64 |
| TSM | ResNet Features | .73 |
| LSTM Attn | Inferred Atomic Events | **.75** |
| Neural TLP | Inferred Atomic Events | .69 |

Table 2: mAP scores versus video baselines.

| Model | $\bar{\mathbf{y}} = 53$ (orig) | $\bar{\mathbf{y}} = 40$ | $\bar{\mathbf{y}} = 30$ | $\bar{\mathbf{y}} = 20$ | $\bar{\mathbf{y}} = 10$ |
|---|---|---|---|---|---|
| LSTM Attn | **.75** | .37 | .34 | .31 | .27 |
| Neural TLP | .69 | **.49** | **.47** | **.45** | **.42** |

Table 3: We observe mAP performance when testing on out of distribution data with respect to the average number of labels per sample.

We experiment where the atomic events are obtained from a noisy environment or inferred from a process upstream, such as our baseball video. To infer the atomic events, we detect the objects through a Faster R-CNN (Ren et al., 2015) and use its cropped image feature and optical flow to predict the shape and movement. These atomic events are predicted and cached prior to inputting them in our models.

From the results in Table 1 we see that our more structured method is the most optimal for extracting rules. MAP provides a coarse representation of the labels and enumerated rules with no parameters. It can express these rules better than LSTMs but lags behind our method. As the number of free parameters increases, the LSTM models are more likely to pick up spurious signals in the data that are useful from a cross-entropy optimization perspective but deviate from the underlying generative rule representation (Hits). This can be seen with highly parameterized LSTM models and video models: I3D (Carreira & Zisserman, 2017), TSN (Wang et al., 2016), and TSM (Lin et al., 2019), leading to larger gains in mAP in Table 2.

Even for mAP, we show that our underlying rule representation is useful when generalizing out of distribution. Here we fix our trained LSTM and Neural TLP models and test on out of distribution data where the frequency of labels is changed in Table 3. We show that Neural TLP performance degrades gracefully as it is exposed to out of distribution data.

We further ablate our Neural TLP model architecture and hyperparameters in D.2. We show the effectiveness of our method on recovering longer dynamic length rules over Temporal MAP in Appendix D.3. Now that we tested Neural TLP on synthetic tasks to empirically verify the rule accuracy, we explore its capabilities on real-world healthcare data.

### 4.2 HEALTHCARE DATA

We test the rules recovered from Neural TLP on patient data in MIMIC-III (Johnson et al., 2016). We specifically look at 2023 patients admitted for sepsis (severe infection) and recover the rules corresponding to stable vitals. This is done through predicting urine output as the composite event, an auxiliary variable indicative of the state of the patient's fluids and circulatory system (Komorowski et al., 2018). There are $|\mathcal{X}| = 82$ different atomic events, composed of drugs administered and patient vitals. The vitals are made into boolean events through logic rules provided by doctors, which

| Model | #@50 | MRR |
|-------|------|-----|
| Neural TLP | **3** | **.04** |
| Temporal MAP | 0 | 0 |

Table 4: We compute the number of relevant rules in top 50 (#@50) as well as the mean reciprocal ranking (MRR) of the correct rules.

| Model | Urine Output mAP |
|-------|------------------|
| Logistic Regression | .75 |
| LSTM Attn (L) | .74 |
| Neural TLP | **.77** |

Table 5: Inference results for the urine output task.

indicate the vital severity: low, normal, or high. The model's task is to learn rules corresponding to normal urine outflow. We present the top rules from Neural TLP and MAP to doctors for verification.

From the results in Table 4, we see this is a difficult problem due to a large number of atomic events and small sample size. However, it indicates that the learning done by Neural TLP is useful to learn and to rank important predicates before rule training. Temporal MAP fails at this task with the increased number of atomic events and longer timelines, which led to many grounded predicates (2-3k) per sample. Therefore the $\mathbf{W}$ weights are very coarse and filled with common relations. In addition to relevant rules, Neural TLP maintains good inference performance (Table 5), which is also an important metric for this domain.

From the doctors' feedback, we also present the rules learned in Appendix E, but emphasize that these are *observed facts*. This means that the rules are partially explained by a subset of predicates and not considered treatment rules, as the rules didn't contain any drugs administered. The doctors did confirm that we captured important explanatory variables in our proposed rules, so we are optimistic about using our framework for feature selection in complex temporal environments.

## 5 RELATED WORK

**Structured Temporal Prediction**    To optimize over the composite events given observed atomic events, Hidden Markov Models (HMMs) are commonly used to model the latent rules that emit consistent atomic events to induce the composite events. Variants of HMMs have been developed to handle symbolic atomic events (Mutschler & Philippsen, 2012; Kersting et al., 2006; Liu et al., 2017) to more perception-based events, such as videos (Tang et al., 2012). Deep models have shown to incorporate temporal logic constraints within their outputs by leveraging Transformers architectures (Finkbeiner et al., 2020), knowledge distillation (Ma et al., 2020), and by representing temporal logic as a differentiable loss (Innes & Ramamoorthy, 2020). However, interpreting latent representations of deep models in order to extract explicit rules is still being researched (Arras et al., 2019; Chefer et al., 2021; Lal et al., 2021).

**Rule Representations and Inference**    To explicitly induce rules over data, a space temporal logic rules can be searched to determine appropriate rules. Temporal rules are induced using satisfiability (SAT) based methods (Neider & Gavran, 2018; Camacho & McIlraith, 2019) given strong priors to the rule structure and limited data noise. In the presence of noise, softening logic rules using Markov Logic Networks (Richardson & Domingos, 2006; Song et al., 2013) or probabilistic logic (ProbLog) (De Raedt et al., 2007; Kimmig et al., 2011) perform approximate inference well, but are intractable for rule training. Recently Yan & Julius (2021) proposed a network to learn sparse weights for temporal rules given a fixed rule format.

**Inductive Logic Programming**    Extracting expressive rules is explored through Inductive Logic Programming (ILP) methods (Muggleton, 1991; Muggleton & De Raedt, 1994) and within Statistical Relational Learning literature (Koller et al., 2007; De Raedt & Kersting, 2010). Given fixed background knowledge, parameterized models softly select the relevant facts used to derive the labels and therefore lift logic rules (Yang et al., 2017; Evans & Grefenstette, 2018). Atoms can also be represented as latent vectors to provide probabilistic facts for increased generalizability (Rocktäschel & Riedel, 2017). Dong et al. (2019) present a more neural architecture to represent first-order logic and scale to larger rule search spaces.

## 6 Conclusion

Composite event extraction is a common task across many temporal domains. It is important to understand the underlying atomic events and their predicates that induce the composite event. We propose Neural TLP that learns these composite event rules even when provided noisy temporal data. It first learns parameters for atomic event timeline compression and pairwise predicate prediction. Then once grounded predicates are reliably inferred, the structure learning stage learns over the space of combinatorial predicates to induce the final rules. We verified our method on synthetic video tasks and explored rule recovery in a real-world healthcare dataset.

**Ethics Statement**   Neural TLP, like any data-driven model, has potential societal impacts that could include extracting unfairly biased rules or relations. In such cases, we envision using such a framework for *knowledge discovery* over end decision making. We operate on hospital patient data, through MIMIC-III which has been de-identified to avoid leaking privileged patient information.

**Reproducibility Statement**   We provide the entire model code for our framework in the supplementary materials. The dataset for CATER is publicly available, and we have provided the processing code to generate our custom CATER data. We are in the process of sharing the experimental code for the MIMIC-III experiments as well. We further describe our optimization procedures (Appendix B), hyperparameters (Appendix D.2), and baseline model architectures (Appendix C) for clarity.

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

## A    TEMPORAL ILP COMPARISON

**Neural LP**    In typical ILP problems, such as Neural LP (Yang et al., 2017), the rules are induced over a static knowledge base. This involves learning the walks along the static graph $\mathcal{G}_S = (\mathcal{E}_S, \mathcal{R}_S)$. Starting at entity node $e_x \in \mathcal{E}_S$ Neural LP learns the associated edge relations $\mathcal{R}_S$ to traverse in order to reach the corresponding ending entity node $e_y$. Since between two entities there can be many arbitrary paths, given the data, the most likely path is found. This is done by learning the attention $\alpha$ over relational predicates matrices $\mathbf{M}_{\mathbf{R_k}}$ to traverse from $\mathbf{v_x}$ to $\mathbf{v_y}$. Here $\mathbf{v_x}, \mathbf{v_y}$ are one hot embeddings of the start and entities respectively.

$$\hat{\mathbf{v}}_{\mathbf{y}} = \mathbf{v_x}^\top \prod_{t=1}^{T} \sum_{k}^{|\mathbf{R}|} \alpha_t^k \mathbf{M}_{\mathbf{R_k}}$$

Then the objective is to maximize the score $\hat{\mathbf{v}}_{\mathbf{y}}^\top \mathbf{v_y}$ of selecting the correct end entity $e_y$ through paths selected by $\alpha_t^k$. The edges along this path compose the predicates of the rule between $e_x, e_y$. Refer to the original paper for full details and implementation (Yang et al., 2017).

**Knowledge Representation**    In temporal rule learning one can also compose a graph between the entities (events) on the timeline. However the structure of such a dynamic graph $\mathcal{G}_D = (\mathcal{E}_D, \mathcal{R}_D)$ over time is different. The dynamic graph typically contains fewer events $|\mathcal{E}_D| < |\mathcal{E}_S|$, and the graph is dense as every event has some temporal relation with respect to all other events. The edge relations may not be provided in unified knowledge base, where the relations can vary per sample $m$: $\mathcal{R}_D = \{\mathcal{R}_D^1, \mathcal{R}_D^2, \ldots, \mathcal{R}_D^m\}$. Furthermore these temporal relations between samples are rarely annotated with the relations of interest, while in static case, the relations $\mathcal{R}_S$ are usually predetermined.

**Rule Induction**    When learning rules in the temporal case, many rules don't conform to the chain like rule structure. This can be seen in the form $f := \texttt{after(a, b)} \land \texttt{before(c, d)}$ where there is no path between the first predicate and the second (see the last rule in Table 9 as an example). Without a path between two events to guide the rule construction, the rule can potentially involve any events and relations. Formulating the problem in terms of a walk is challenging to evaluate if we consider all relations between the disjoint events and selecting the most likely rule:

$$\arg\max_{r} \sum_{p_i \in \mathcal{P}} \texttt{after(a, b)} \land \texttt{before(c, d)} \land p_i(X, Y) \quad \forall X, Y \in \text{disjoint}(\texttt{(a, b)}, \texttt{(c, d)})$$

Here, events between two pairs of predicates are disjoint if there does not exist a common event between the two predicates. If not, we would have to sample all combinations of $X, Y$ from each predicate respectively: $\texttt{(a, c)}, \texttt{(a, d)}, \texttt{(b, c)}, \texttt{(b, d)}$. Instead of expensive marginalization, we aim to search for combinatorial combinations of grounded predicates in a differentiable fashion. This search space for Temporal ILP grows faster with larger numbers of atomic events and relations.

**Lemma.**    *In Temporal ILP, given the space of events $E = |\mathcal{X}|$, predicates $P = |\mathcal{P}|$, and a rule with $n$ predicates, the search space for a rule $r$ is generally $\mathcal{O}((PE^2)^n)$. If $\mathcal{P}$ consists of symmetric relations, such as $\texttt{before}(u, v) = \texttt{after}(v, u)$, and an equivalent relation $\texttt{during}(u, v) = \texttt{during}(v, u)$, we keep $\lfloor \frac{P}{2} \rfloor$ of the symmetric relations and the unique values in the equivalent relation. Then the bound can be tightened to $\Theta((\lfloor \frac{P}{2} \rfloor E^2 + \frac{E(E+1)}{2})^n)$ unique combinations of rules.*

*In ILP given the knowledge base, to construct a rule for predicates with a pair of entity variables $E_1, E_2$ and average node degree $D$, the search space is $\Theta(D^n)$.*

The corresponding temporal models and rule extraction methods has to reflect these differences. We focus on structured time series tasks where we have to learn temporal predicate parameters in addition to the temporal logic formulas from the samples. Our model Neural TLP learns these temporal relations between pairwise atomic events, with considerations of event noise as well as computational complexity. Then given the inferred relations, the combinations of relations can be learned such that the correct composite event is inferred.

# B  OPTIMIZATION

We used the Adam (Kingma & Ba, 2014) optimizer with a learning rate of 0.001 for all our experiments and the default parameters described in their paper. The batch size during relational training is set to 256. Each experiment was run for 100 epochs to train the relation parameters and weights $\mathbf{W}$. For variable rule length search we tested longer epoch lengths, but didn't see much improvement in the validation past the first epoch. Therefore 1 epoch of tuning was done to optimize the combinatorial attention weights $\mathbf{s}$ while freezing all other relation parameters. All experiments were conducted on a server with a Nvidia 2080TI GPU with 11GB of VRAM.

We made sure all our model configurations could fit on a single GPU of this size. The memory intensive component came from having different attention weights $\mathbf{s}$ and combinatorial matrices $\mathbf{C}$ for each rule $r \in \mathcal{R}$ learned. This meant that we had to limit the number of predicates combinations we search over. Here $c$ is the number of most relevant predicates searched per rule and $n$ is the max number of predicates, so the number of combinations for the max rule length is $\binom{c}{n}$. So for each variable rule length of 1,2,3,4 we chose $c = 100, 100, 30, 25$ respectively. Due to the larger memory requirements, we reduce the batch size during rule search to 64.

# C  BASELINES

## C.1  LSTM

Since we are working with multi-hot labels, and co-occurring atomic events, it is difficult to parametrize existing HMM variants. Therefore we test a more neural recurrent baseline, LSTM (Hochreiter & Schmidhuber, 1997), over the raw atomic event stream. To account for object invariance, we marginalize over the timelines to produce the inputs $\tilde{\mathbf{M}}_{\mathbf{T}} = \sum_k \max(1, \mathbf{M}_{\mathbf{T}k,:,:}) \in \mathbb{R}^{|\mathcal{X}| \times T}$. The compressed timelines and temporal indexing are done over $\tilde{\mathbf{M}}_{\mathbf{T}}$ to produce $\mathbf{M}_{\mathbf{A}}$. From $\mathbf{M}_{\mathbf{A}} \in \mathbb{R}^{|\mathcal{X}| \times t}$ we input $\mathbf{x} \in \mathbb{R}^{|\mathcal{X}|}$ at each step $t$: $\mathbf{x}_i, \mathbf{h}_i, \mathbf{c}_i = \mathrm{LSTM}(\mathbf{x}_{i-1}, \mathbf{h}_{i-1}, \mathbf{c}_{i-1})$, where $\mathbf{c}$ is the cell state. Then we perform classification over the labels through a linear layer using the last hidden state $\hat{\mathbf{y}} = \mathbf{W}\mathbf{h}_t + \mathbf{b}$. We test both a large (L) and a small (S) version with hidden dimensions of $|\mathbf{h}| = 512$ and $|\mathbf{h}| = 160$ respectively.

We also test an attention based variant where the attention value for each hidden state is computed using the hidden state as well as the input atomic events at that step. This helps the model focus on time steps that are not empty over long frame sequences, and led to better convergence.

$$
\begin{aligned}
\mathbf{r}_i &= [\mathbf{h}_i; \mathbf{x}_i] \\
a_i &= \tanh(\mathbf{b}_1^\top (\mathbf{r}_i \mathbf{W}_1)) \\
\mathbf{a} &= \mathrm{softmax}(\mathbf{a}) \\
\mathbf{h} &= \sum_i \mathbf{h}_i \cdot a_i \\
\hat{\mathbf{y}} &= \mathbf{W}_2 \mathbf{h} + \mathbf{b}_2
\end{aligned}
$$

Here $\mathbf{W}_1 \in \mathbb{R}^{(|\mathbf{h}|+|\mathbf{x}|) \times d}$ and $\mathbf{b}_1 \in \mathbb{R}^d$ project the concatenated data into attention dimension $d$, which was set to $d = 32$ for all our experiments.

To extract the rules, we first enumerate all possible composite events. Then we synthesize each composite events as raw event timeline data. Unlike the original training data where there are multiple atomic events occurring simultaneously, for each composite event we have two atomic events occurring unambiguously `before`, `during`, or `after` one another. Then we pass the synthesized events into the model and take the argmax prediction as the corresponding rule label.

## C.2  TEMPORAL MAP

While LSTMs optimize for label prediction, a baseline to test rule induction is to maximize the posterior distribution of the enumerated composite event rules given the training labels. This is done by using Neural TLP but with two changes. First we freeze all model parameters as done

in a deterministic setting. Second, instead of learning the attention weights $\mathbf{W}$ we count the co-occurrences of relations computed through $\mathbf{M_R}$ and the training labels for each sample.

To compute $\mathbf{W}$, we start with all the observed time series samples $\mathbf{T}, \mathbf{Y} = \{\mathcal{T}\}_{i=1}^m, \{\mathbf{y}\}_{i=1}^m$. Instead of operating over $\mathbf{T}$ we are interested in the grounded relational data $\mathbf{R} = \{\mathbf{M_R}\}_{i=1}^m$, which can be extracted through the prior stages of the Neural TLP pipeline through the TLN network $g_\theta$ with fixed relations, as described in 3.1.

Given the inferred $\mathbf{M_R}$ data, we have pairs of $\mathbf{M_R}^k, \mathbf{y}^k$ for each sample of $k \in [1, m]$. Then we compute the co-occurrences of grounded predicates $p$ and labels $y$ as:

$$\Theta_{i,j} = \sum_{\mathbf{M_R}^k, \mathbf{y}^k} \mathbb{1}_{p_i \in \mathbf{M_R}^k, y_j \in \mathbf{y}^k} \quad \forall k \in [1, m]$$

$$\hat{\mathbf{W}}_{i,j} = \frac{\Theta_{i,j}}{\sum_j \Theta_{i,j}}$$

This $\mathbf{W}$ is used for rule learning in the same manner as Neural TLP for both fixed and variable length strategies. To compute mAP we always predict the top $\bar{\mathbf{y}}$ most frequently occurring labels in the training set.

# D CATER

We explore composite action prediction over complex videos in the CATER data set (Girdhar & Ramanan, 2019). The CATER data set provides synthetic videos of multiple objects performing different actions simultaneously over the duration of the videos. The atomic events are a conjunction of these movements $\in$ {rotate, slide pick-place, contain} and objects $\in$ {cone, cube, sphere, snitch}. Such an atomic event is slide cone $\in \mathcal{X}$ where $|\mathcal{X}| = 14$ and temporal predicates $r \in$ {before, during, after}.

The composite event rules are composed of a single grounded temporal predicate ($n = 1$) between two atomic events. For example the underlying composite events in the video such as before(pick-place cube, rotate snitch) is assigned to label 2, providing $|\mathbf{y}| = 301$ unique composite events. Only the label along with the videos are provided during training, while the rules are unknown.

Furthermore all these atomic events occur randomly during the video, thus multiple labels corresponding the the underlying temporal rule are active. This provides a difficult, yet practical challenge: we know what coarse composite events occur during a timeline, but we want to recover the underlying atomic events and predicates (rules) leading to each of these composite events (labels) as shown in Figure 5. Since the videos are generated with underlying rule templates, we can objectively test our models to recover these rules. For our experiments we used the train, validation, and test splits provided in original dataset.

## D.1 PREDICTED ATOMIC EVENTS

| Modality | Overall Acc | Rotate | Slide | Pick-Place |
|---|---|---|---|---|
| RGB | 85.7% | 17.6% | 3.9% | 91.3% |
| RGB + Flow | 96.6% | 76.9% | 88.2% | 96.9% |

Table 6: Here are the accuracies for the predicted atomic events. The results show accuracy predicting the event only using the RGB image features and with the optical flow information. Further rules were used to predict contain.

Using the original CATER video data we performed experiments where the atomic events were provided and where we inferred the atomic events. The latter case is more difficult, yet more realistic for rule recovery over collected data where noise exists. To infer the atomic events we first tune a Faster R-CNN (Ren et al., 2015) over the object bounding boxes. In real world use cases it may

| Loss | Relation Parameters | Projection | mAP | Rules@1 | Rules@5 |
|------|---------------------|------------|-----|---------|---------|
| $\mathcal{L}_{ent}$ | $T = t = 301$; no conv | ✗ | **.693** | .830 | **.966** |
| $\mathcal{L}_{ent}$ | $t = 150$; kernel= $14 \times 3$, stride= 2 | ✗ | .682 | .883 | .953 |
| $\mathcal{L}_{ent}$ | $t = 50$; kernel= $14 \times 7$, stride= 6 | ✗ | .679 | .873 | .960 |
| $\mathcal{L}_{ent}$ | $t = 150$; freeze conv | ✗ | .669 | .869 | .953 |
| $\mathcal{L}_{ent}$ | $t = 150$; freeze $\alpha$ | ✗ | .656 | .671 | .950 |
| $\mathcal{L}_{ent}$ | $t = 150$; freeze $\beta, \gamma$ | ✗ | .658 | .681 | .913 |
| $\mathcal{L}_{ent} + \mathcal{L}_1$ | $t = 150$ | ✗ | .681 | **.915** | .958 |
| $\mathcal{L}_{ent} + \mathcal{L}_1$ | $t = 150$ | ✓ | .675 | **.913** | .956 |

Table 7: We ablate the different components of Neural TLP over the CATER predicted atomic events data. We start by determining the efficacy of quantization, where $t$ is the dimension after quantization. Then we test the contribution of each parameter in relation learning. Finally we test additional optimization strategies.

be possible to use pre-trained detectors to lift these bounding boxes, but they typically don't cover synthetic objects. We use these bounding boxes to generate a visual feature over the cropped image using a pre-trained ResNet-50 (He et al., 2016). We use the image feature and optical flow from the previous and next frame to predict the shape and movement.

There is additional difficulty performing this inference and we present the event accuracy in Table 6. Furthermore, the action `contain` is hard to distinguish from `pick-place`, so additional rules are used to disambiguate `pick-place` from `contain`, leading to additional uncertainty. The rule classified `contain` if the movement is a `pick-place` and the bounding box of the moving object is close to the bounding box of a static object. Since each object has its own atomic actions we assigned a timeline for each object, tracking a max of $k = 30$ objects.

### D.2 MODEL ABLATION

We ablate our model over the predicted atomic event data in Table 7. In the first section we test different convolution kernel sizes which contains a 1d covolution weight that is learned per atomic event $|\mathcal{X}| = 14$ and a stride. We see that convolution smoothing is beneficial at the reduced dimension of $t = 150$ and use this for further ablations.

In the next section we test how much each relation parameter contributes to the result. The convolution weights are fixed to $\mathbf{1}$, and we see that in this case they contribute minimally to rule induction. In real world cases with more complex events, these learned convolutions could potentially be more useful if systematic noise exists. For the convolution scalar $\alpha$ it is crucial to shift the compressed timeline to correctly identify event time intervals, as seen when it is fixed to 1. Similarly when shift $\beta = \mathbf{0}$ and scale $\gamma = \mathbf{1}$, it misses on variations in the relation data.

In the final section we test our $\mathcal{L}_1$ regularization to produce sparser results for $\mathbf{W}$. This shows to enable better rule induction. We test projecting $\mathbf{W}$ between $[0, 1]$ and while we got similar results, we notice that the optimization converges faster and in a more stable fashion. We primarily focused on *rule precision* by looking at Hits@1, while increasing recall produced similar results for all methods in the setting where the rules only have $n = 1$ predicate.

### D.3 DATA ABLATION

To test different rule lengths $n$, training samples $m$, and event complexity $\bar{\mathbf{y}}$, we generated more CATER-like data to explore these dataset statistics. Here we use the same atomic events and relations as in the original dataset. Due to the number of possible combinations of the dataset statistics, it is expensive to generate the video data and run our vision pipeline to generate the predicted atomic events. Instead we opted to simulate the atomic event predictions directly. For each sample we:

1. Randomly sample $j$ rules containing up to $n$ predicates.

| Max Len $n$ | $\bar{\mathbf{y}}$ | Model | Variable | Len 1 | Len 2 | Len 3 | Len 4 |
|---|---|---|---|---|---|---|---|
| 1 | 1.0 | TLP | $.97_{\pm.04}$ | $.97_{\pm.04}$ | - | - | - |
|   |   | MAP | $.53_{\pm.07}$ | $.53_{\pm.07}$ | - | - | - |
| 2 | 4.0 | TLP | $.44_{\pm.08}$ | $.80_{\pm.15}$ | $.09_{\pm.01}$ | - | - |
|   |   | MAP | $.17_{\pm.00}$ | $.34_{\pm.00}$ | $.00_{\pm.01}$ | - | - |
| 3 | 5.3 | TLP | $.15_{\pm.02}$ | $.35_{\pm.06}$ | $.05_{\pm.02}$ | $.01_{\pm.01}$ | - |
|   |   | MAP | $.11_{\pm.00}$ | $.33_{\pm.00}$ | $.01_{\pm.01}$ | $.00_{\pm.00}$ | - |
| 4 | 10.0 | TLP | $.10_{\pm.01}$ | $.36_{\pm.05}$ | $.02_{\pm.03}$ | $.00_{\pm.01}$ | $.00_{\pm.00}$ |
|   |   | MAP | $.09_{\pm.01}$ | $.33_{\pm.04}$ | $.04_{\pm.03}$ | $.00_{\pm.01}$ | $.00_{\pm.00}$ |

Table 8: For $j = 1$ samples we compare model Hits@10 performances across different rule lengths. We observe the total variable length Hits for rules of all lengths that have to be learned. We break down the combined performance into the individual performance per fixed rule length $n = $ Len i, that compose the total variable performance.

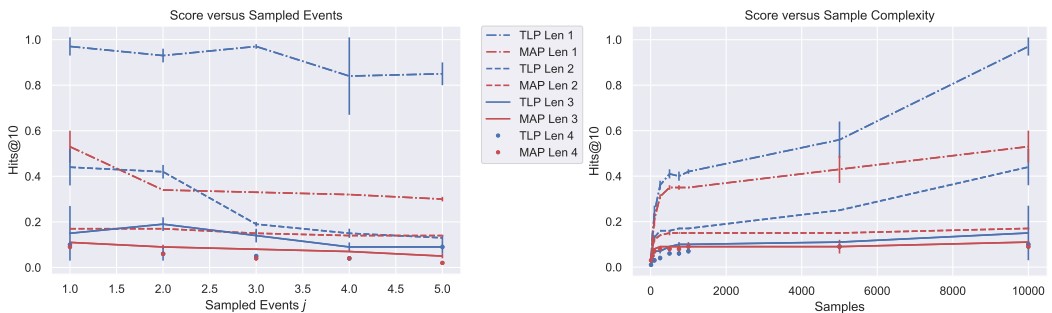

Figure 6: We compare Neural TLP (Blue) and Temporal MAP (Red) with varying number of active labels and samples. Each curve represents the overall variable length accuracy up to rule length $n$. On the left we compare the variable length performance when we sampled more event rules per video, increasing the active labels and timeline noise with fixed 10000 samples. On the right we compare performances as the number of samples increase and fix $j = 1$.

2. Synthesize a timeline for each rule, where atomic events are placed in the timeline consistent with the rule. Each atomic event prediction was sampled from a normal distribution, with means centered around the atomic event detection accuracies in Table 6.

3. Based on the synthesized timeline over sampled rules, add any consistent rules induced by the synthesized timeline. These additional consistent rules contribute to our $j \leq \bar{\mathbf{y}}$ estimate.

4. Gaussian noise was also added along the timeline where events did not occur to simulate detection noise.

Given the synthesized timeline and the consistent rules, the corresponding $\mathbf{M_T}$ and labels $\mathbf{y}$ can be generated for each sample. For validation and testing data we generated 2500 samples each, regardless of the number of training samples.

From this base data set, we first vary the max rule length $n$ of the rule samples and fix $j = 1$ to isolate the effect of the rule length. For every level $n$ we sample 100 total rules up to the max length $n$ for consistency. Breaking down the variable length accuracy in Table 8 we see Neural TLP provides better performance for shorter rules, while rule learning becomes more difficult for longer rules.

We also test the event $j$ and data points $m$ sample complexity in Figure 6. As we sample more events $j$, more noise is added to our timeline and makes it harder for models to recover the underlying rules. Inversely we also show the performance increases with the number of samples, where Neural TLP is more sample efficient.

# E MIMIC-III

```
    after(oral water, spo2_sao2 high) ∧ after(oral water, paco2 high)
after(hco3 high, spo2_sao2 high) ∧ after(spo2_sao2 high, calcium high)
    after(hco3 high, spo2_sao2 high) ∧ before(hco3 normal, pao2 low)
```

Table 9: Induced rules for normal urine, verified as correct or plausible by doctors.

| | |
|---|---|
| oral water | patient drank water |
| spo2_sao2 | pulse oximetry $S_pO_2$ and blood gas $S_aO_2$ in oxygen |
| pao2 | partial pressure of blood oxygen $P_aO_2$ |
| paco2 | partial pressure of carbon dioxide in the blood $P_aCO_2$ |
| hco3 | body metabolic bicarbonate $HCO_3$ |
| calcium | patient calcium indication |

Table 10: Descriptions of MIMIC atomic events in the lifted rules from Table 9.

For the healthcare data we use the MIMIC-III dataset (Johnson et al., 2016). The dataset contains measurements, vitals, and medications for intensive care patients, and is *already de-identified*. We first filter the patients containing ICD codes corresponding to sepsis and sample 2k patients.

Instead of inferring patient survival directly, the medical doctors suggested to look at circulatory indicators, specifically urine flow. Additionally they helped us identify a subset of patient vitals and measurements to use for urine flow, leading to our $|\mathcal{X}| = 82$ boolean atomic events. From the 2k patients we identify timepoints containing urine information. This serves as a label, and all events before the urine event are the timeseries inputs. Since patients had multiple indicators of urine during their stay we have 3.8k samples of time series and urine label (normal or low). During training we created an 80/20 split for training and validation respectively.

After training and iterative feedback from the doctors, we successfully lifted useful indicators of urine flow as presented in Table 9. A description of the atomic events is presented here in Table 10.

