# OpenReview forum: "Neural Temporal Logic Programming"
_ICLR.cc/2022/Conference — ICLR 2022 Submitted_

### Official Review · Reviewer_pPed · 2021-11-01

**Correctness:** 3
**Technical Novelty And Significance:** 3
**Empirical Novelty And Significance:** 3
**Recommendation:** 5
**Confidence:** 3

**Main Review:**

The paper addresses an important problem of learning temporal rules from data, for which many applications exist (as the paper convincingly motivates).

A very strong point of the paper is to learn interpretable rules using modern neural network machinery. This can certainly inspire future work.

The paper is not always easy to follow. It would have been useful to more explicitly state the aims of the proposed method. For example, on page 4 the paper refers to Temporal Relation Networks, but they are dismissed because "timelines can be long in our setting", yet, it is not clear what this refers to. For me, it was also quite unnatural to say that composite events are a-temporal (e.g. in the baseball
example on page 2). Why does it make sense that you cannot say anything about the timeline of the 'strike' event? Also in the remainder of Section 3.1, various choices are made, which were not all obvious to me.

In the structure learning part, I was missing the connection to probabilistic rule learning approaches (e.g. ProbFOIL). The essence
seems to be to learn rules where the temporal atoms have a probability attached to it, unless I am missing something.

In the evaluation, the paper convincingly shows that variations of the proposed method performs less well. What was not convincing to me
were two aspects:
(1) Why previous approaches are unsuitable for the proposed settings, e.g. probabilistic rule-learning, sequence mining, etc.
(2) I have doubts about the validity of the medical example. For example, isn't it the case that 'oral water' is simply responsible for
urine flow rather than after(oral_water, spo2_sao2=high) ? I can understand that a doctor would say that you "captured important explanatory variables" in the proposed rules, but that does not mean that it is in any way true. In fact: doesn't this show a downside of the approach in the sense that only temporal events can be used in rules?


**Summary Of The Paper:**

This paper presents an approach for learning temporal rules from data. The idea is to extract atomic events, and their temporal relationships between them. Subsequently, composite events are learned using composite event labels for supervision. The whole approach is formulated as an optimisation problem, after which standard techniques are applied to solve it.


**Summary Of The Review:**

Overall, I thought this is an interesting contribution to the ICLR conference. I do have some doubts about the completeness and validity of the experiments.

---

> ### Author Response · Authors · 2021-11-15
> **Respose to Reviewer pPed (3/3)**
>
> ### Comparison to Rule Mining
>
> **In the structure learning part, I was missing the connection to probabilistic rule learning approaches (e.g. ProbFOIL). The essence seems to be to learn rules where the temporal atoms have a probability attached to it, unless I am missing something. Why previous approaches are unsuitable for the proposed settings, e.g. probabilistic rule-learning, sequence mining, etc.**
>
> At a high level we produce intermediate probabilistic predicates after parameter learning and learn probabilistic rules during structure learning, but there are key differences when learning temporal rules.
>
> In ProbFOIL and many rule learning approaches, rules are of a chain structure in its path finding process. Thus such methods learn rules over graph structured data, where the rule is the correct path from source to target nodes. However such rules are only a subset of rules in the temporal case, where we have multiple disjoint subgraphs. For example,  make sandwich := before(slice bread, toast) ^ before(wash tomatoes, chop). Here there is no connection between the two predicates, so one predicate can happen before, during or after the other. We have to take into consideration a much larger rule space of disjoint clauses, while it is not clear how such rule learners can adopt this rule space.
>
> Furthermore in the temporal case we are not provided source and target terms in the labels. Therefore we consider all atomic events present in each sample as candidate preidcates for our rules, following our strike example from earlier. So with temporal rules we have a larger space of disjoint rules, but we also have to *search* over this entire space of rules.
>
> Another distinction is that ProbFOIL uses accuracy heuristics to determine thresholds for when to stop the rule learning by leveraging both global and local rule accuracy for new clauses to add. In our case we only optimize a single loss to score the combination of predicates used to infer each label end-to-end. Furthermore such neural frameworks lets us optimize all rules in a joint fashion, which is beneficial with a large number of composite events. These provide better practicality for rule learning in our setting.
>
> Of course such practicality is seen in neural approaches as well such as neural inductive programming methods, Neural LP [1], $\partial \text{ILP}$ [2], Neural Theorem Provers [3], to name a few. However these also focus on path finding for rule learning, and it is unclear how to address the disjoint problem in the temporal setting. This inspired us to build Neural TLP on top of these ideas. We provide more details of the temporal search space and the comparison to these logic programming methods in Appendix A.
>
> ### Doubts Regarding Medical Results
>
> **I have doubts about the validity of the medical example. For example, isn't it the case that 'oral water' is simply responsible for urine flow rather than after(oral_water, spo2_sao2=high)?**
>
> In the medical setting we are trying to demonstrate a real world use case where the data is inherently noise and there are many atomic events to consider. Yes, it is reasonable that urine flow is affected by oral_water, in conjunction with other hidden factors. Here we try to illustrate that a model should be able to recover such combinations of events and propose rules in such a complex setting.
>
> This complexity is evident with Temporal MAP, which **did not** have the oral_water event in *any* of its top 50 rules, since it only choose common relations in the timeline like high blood pressure. In comparison our method can attend to the most useful atomic events to learn such predicates.
>
> **I can understand that a doctor would say that you "captured important explanatory variables" in the proposed rules, but that does not mean that it is in any way true. In fact: doesn't this show a downside of the approach in the sense that only temporal events can be used in rules?**
>
> Yes, we emphasize that these proposed rules are correlations with urine flow as best understood by the doctors. Of course it would take a more thorough study to understand if these rules truly lead to the urine output.
>
> While we focus on the temporal rules, it makes sense that there are also other a-temporal rules leading to the outcome. For example in the medical case we could consider the patients medical history, family illnesses, genomic features, etc. in addition to their online temporal vitals to construct rules for urine flow. Such joint temporal and static rules would be more comprehensive for future work, thanks for the comment!
>
> ### Closing Comments
> Thank you for giving us an opportunity to clarify our work. Please let us know if you have any further questions.
>
> [1] Yang et al. Differentiable Learning of Logical Rules for
> Knowledge Base Reasoning. NeurIPS'17
>
> [2] Evans and Grefenstette. Learning Explanatory Rules from Noisy Data. JAIR'18
>
> [3] Rocktäschel and Riedel. End-to-End Differentiable Proving. NeurIPS'17

---

> ### Author Response · Authors · 2021-11-15
> **Respose to Reviewer pPed (2/3)**
>
> **Also in the remainder of Section 3.1, various choices are made, which were not all obvious to me.**
>
> We can start by clarifying the high level steps going on in 3.1. First we take the raw timeseries data per atomic event and smooth it through convolution. Then we propose differentiable method to select the start and end times within atomic events. These intervals are used to compute pairwise predicates between atomic events across all objects. Finally these predicates are further used to infer the composite event labels. Now let's dive into certain design choices that might lead to confusion.
>
> 1. Scalar $\alpha$: This is used to control the magnitude of the score. This is done since we need to determine a threshold $\epsilon$, where above this we say the atomic event occurs and below we say it does not occur. Say for an atomic event we have a score .2 for the timepoint where the ground truth atomic event is taking place. If we set our $\epsilon=.22$ then we will determine that the event does not occur at that timepoint. Here if we learn $\alpha=1.5$ in our end-to-end pipline, it enables to push our point $.2 * 1.5 = .3 > .22$ above the threshold to be classified correctly. Simultaneously we are learning $\alpha$ to push noisy points below $\epsilon$. This logic is necessary in computing the event interval start, described next.
> 2. Event start (Eq 3,4): We recommend reviewing Figure 3 to understand how this is computed. Essentially we leverage our $\epsilon$ to mask out all times where the event does *not* occur by setting it to the max score observed in the timeline. Then the remainder of the timeline contains occuring event scores, increasing over time. To compute the start, we simply take the min of these scores.
> 3. During score (Eq 7): Given the start and end times of two atomic events we can easily compute before and after. For during, it can be logically expressed as event v ends after u starts $v_{end} - u_{start}$ **and** u ends before v starts $u_{end} - v_{start}$, indicating an overlap between u and v. Lets start with an example: v starts at time 20 and ends at 25, u starts at 10 and ends at 22. Then we would have our *unnormalized* during score as $min(\{25 - 10, 22 - 20\}) = 2$, whereas if any of these conditions failed the score would be $\leq 0$.
> 4. Predicate parameters $\beta, \gamma$: These parameters enable scaling and shifting for each before, during and after scores to account for noise. A use case is if there is noise in our timeline where the model is overpredicting during, but it is really before or after. Then the scale and shift can reduce the incoming during scores, such that only confident during predictions are passed through.
> 5. Predicate suppression (Eq 8,9): The above predicate scores will only be computed correctly if there are two active atomic events being compared. Say due to noise event u has as start and end of 23 due to some noisy datapoint, even though the event does not occur. Normally our method would predict $min(\{25 - 23, 23 - 20\}) = 2$ and say it is a during relation. Since we know that there are some possiblities of noise for events that do not occur, we can check for these by computing the interval of these events (Eq 8). If the interval is small, then we dictate that the atomic event does not occur, and do no make any predicate predictions with that event (Eq 9).
>
> We demonstrate many of these choices in our system where we ablate each component in Appendix D.2, Table 5. If you had specific choices you wanted to clarify, we are happy to futher comment.

---

> ### Author Response · Authors · 2021-11-15
> **Respose to Reviewer pPed (1/3)**
>
> We thank the reviewer for their thoughtful response. Here we address the questions directly.
>
> ### Method Discussion
>
> **For example, on page 4 the paper refers to Temporal Relation Networks, but they are dismissed because "timelines can be long in our setting", yet, it is not clear what this refers to.**
>
> In Temporal Relation Networks they leverage sampling to select 2-7 timepoints within the entire timeline to compute relations from. When we look at the videos in CATER, there might be an object movment that happens for 15 frames out of the 300 frames of the video clip. If we randomly sampled which atomic event scores to use, then we might miss these critical frames entirely. Generally with longer timelines, if atomic events occur for short periods of time then we need a method that can robustly capture these event intervals.
>
> Hence we use the architecture described in Figure 3 and Equations 3 and 4 to select these start and end points directly. This interval is computed in a differentiable fashion which enables to pass gradients to prior convolution layers in Equation 2.
>
> **For me, it was also quite unnatural to say that composite events are a-temporal (e.g. in the baseball example on page 2).**
>
> To clarify here, we are saying that these baseball composite events are temporal. Since these composite events such as steal happen in a certain order: run before pitch. Here in Figure 1 we visualize these temporal relations as edges between atomic events.
>
> **Why does it make sense that you cannot say anything about the timeline of the 'strike' event?**
>
> When we are looking at these timelines, for example baseball videos, they are typically contain many atomic events and potentially multiple composite events as well. For example in a baseball highlight video we can observe players pitching, swinging, hitting, missing, and running and might be provided with the composite event labels of strike and steal. Given these atomic events, which atomic events should be used for strike, or for steal? Of course one familiar with the game will tell you for strike you will need a pitch, a swing, and then a miss.
>
> However, in domains where such underlying rules are unknown and we rely on the model to select these atomic events, this becomes non-trivial. Therefore our model considers pairwise relations between all atomic events when composing temporal rules for each composite event.

---

### Official Review · Reviewer_9nKB · 2021-11-02

**Correctness:** 3
**Technical Novelty And Significance:** 2
**Empirical Novelty And Significance:** 1
**Recommendation:** 3
**Confidence:** 3

**Main Review:**

Strengths: the two-step approach (i.e., parameter learning followed by structure learning) does seem novel for the problem.

Weakness: the empirical results are rather unconvincing. Not only that the baselines (two LSTMs) are rather weak, but also that the proposed approach seems to be even worse than the baselines in terms of mean average precision (which is the mainstream metric used on CATER) in Table 1. Besides, Table 3 suggests that the proposed approach is marginally better than logistic regression.

Minor:  In figure 2, “(1) parameter learning” seems to be in the wrong place. That should refer to predicate prediction according to page 3, paragraph 1.

**Summary Of The Paper:**

The paper proposed a new approach called neural logic programming to recover composite events from complex time series data using atomic events and their temporal relations.

**Summary Of The Review:**

Although there is some novelty in the proposed approach, the empirical results are too weak to validate the effectiveness of the approach. I do not think the paper in its current form is good enough to be published in ICLR.

---

> ### Author Response · Authors · 2021-11-15
> **Respose to Reviewer 9nKB**
>
> We thank the reviewer for their feedback. We directly address points made by the reviewer.
>
> **Not only that the baselines (two LSTMs) are rather weak, but also that the proposed approach seems to be even worse than the baselines in terms of mean average precision (which is the mainstream metric used on CATER) in Table 1**
>
> Ideally a model will have the correct underlying representation (Hits) and best inference performance (mAP), however our results show that this is hard to achieve. A highly paramterized LSTM might learn spurious correlations in the data, but still get the correct label. This leads to higher mAP, but close to 0 rule recovery performance.
>
> We test this hypothesis by using the CATER rules to synthesize new test timelines containing fewer average composite event labels per sample $\bar{\mathbf{y}}=[10, 40]$. This gives us a different distribution from our original training data where $\bar{\mathbf{y}}=53$. This shows a situation where the data distribution might shift $\bar{\mathbf{y}}=[10, 40]$ and we only have models trained on the source distribution $\bar{\mathbf{y}}=53$. Looking at test mAP on these target distributions, Neural TLP's mAP performance degrades more gracefully than LSTM as the target test data shifts away from the training source data. This shows the power of the underlying inductive baises in our model for generalization.
>
> | Model          | mAP $\bar{\mathbf{y}}=53$ (orig) | $\bar{\mathbf{y}}=40$ | $\bar{\mathbf{y}}=30$ | $\bar{\mathbf{y}}=20$ | $\bar{\mathbf{y}}=10$ |
> |----------------|------------------------------------------|-------------------------------|-------------------------------|-------------------------------|-------------------------------|
> | Neural TLP     | .69                                      | **.49**                       | **.47**                       | **.45**                       | **.42**                       |
> | Attention LSTM | **.75**                                  | .37                           | .34                           | .31                           | .27                           |
>
> We also test a setting where Neural TLP focuses on optimizing mAP, where we change our mapping function from grounded predicates to the labels. Previously we defined this function $f_\phi$ as logistic regression, where we interpreted relevant predicates from its weights $\mathbf{W}$. We add another head to our model $f_\Theta=$ MLP which is a dense network primarily focused on optimizing for mAP given the predicted predicates. With this head we increase our mAP from .69 to .72. This indicates that we can add additional parameters if our only goal is mAP.
>
> **Besides, Table 3 suggests that the proposed approach is marginally better than logistic regression.**
>
> In the medical domain, model performance as well interpretability are very important. For rule interpretability we show that our model is able to better select the relevant predicates and atomic events for the rules than Temporal MAP. For mAP we are on par with LSTM performance, but we know that LSTMs learn spurious latent rules from the CATER experiments, thus harder to trust in such critical domains.
>
> **In figure 2, “(1) parameter learning” seems to be in the wrong place. That should refer to predicate prediction according to page 3, paragraph 1.**
>
> When we refer to parameter learning, we meant the process of learning *all* parameters needed for predicate prediction end-to-end. This includes the parameters for convolution $\mathbf{K}, \alpha$, predicate parameters $\beta, \gamma$, and projection of predicates $\mathbf{W}$ to the label space. We agree that this paragraph was not clearly written to reflect this, so thank you for bringing this to our attention!
>
> ### Closing Comments
> Thank you for giving us an opportunity to clarify our work. Please let us know if you have any further questions.

---

### Official Review · Reviewer_X4hj · 2021-11-02

**Correctness:** 3
**Technical Novelty And Significance:** 3
**Empirical Novelty And Significance:** 3
**Recommendation:** 5
**Confidence:** 3

**Main Review:**

**Strengths**

- The paper is well written.
- The proposed solutions, namely the differentiable layers to predict the temporal relations between atomic events and the post-hoc strategy to induce propositional rules from the last linear layer, are simple but non trivial.
- Results are promising.

**Weaknesses**

- It is not clear how the work positions with respect to recent literature in spatio-temporal reasoning and video question answering. For example, the authors should have a look at [1]-[2]
- While experimental results are interesting and promising, they provide only a partial picture, as they neglect recent neural advances. Indeed, comparisons are made against only simple baselines and should be conducted against more “structured” neural approaches, see for instance graph-based approaches [1]-[2], where code is available.

**Questions**

1. How scalable is the proposed strategy in terms of number of objects, number of atomic events and compositionality of the target events?
2. As far as I understand, the proposed strategy considers only the temporal relations among atomic events and objects are considered independent of each other. Is that correct? If so, why is that the case?
3. While I understand the intuition about the strategy to extract the time intervals of atomic events (as shown in Figure 3), I do not see how well this strategy can perform in the continuous and more noisy setting. How reliable are the obtained estimates?
4. In CATER experiments, is the perception component (i.e. Faster R-CNN) trained jointly to the Neural TLP  or are two components trained separately?
5. The names parameter and structure learning are a bit confusing in this context. In their classical definition, parameter learning considers learning the parametric relations among variables in a given graph. In Neural TLP, the temporal relations are not given, rather learnt. Why is this considered as parameter learning?
6. Can you mention something about the logic expressiveness of the induced rules?

**Summary Of The Paper:**

The paper proposes an end-to-end differentiable strategy (called neural TLP) to learn unknown temporal relations between atomic events (like after(miss, swing), “miss occurs after swing” in the baseball example), subsequently used to predict composite events (like strike). The strategy consists of a cascade of a smoothing stage to filter  out noise from the input time series, an interval time extractor, a stage predicting temporal relations (such as before, after and during) and a linear output layer. Furthermore, the paper proposes a post-hoc procedure to extract propositional logical rules relating atomic events to composite ones from the last layer. The performance of the proposed strategy is evaluated on video recognition (CATER) and healthcare (MIMIC-III) datasets against two baselines, namely a LSTM neural net and a simplified version of the proposed strategy.

**Summary Of The Review:**

The paper is clearly written and easy to follow. I like the idea around the proposed differentiable operators used to predict the temporal relations among atomic events. Also, the experimental results are interesting and promising. However, they are preliminary. I would like to see more in-depth analysis on the correcteness of the time intervals for atomic events, the scalability of the proposed strategy and a comparison against recent structured neural approaches for spatio-temporal reasoning. Furthermore, the authors should discuss and relate their work against existing literature in the field of video question answering.

---

> ### Author Response · Authors · 2021-11-10
> **Citations Confirmation**
>
> We thank the reviewer for their feedback, and will be addressing the questions point by point soon. In the review we see citations listed as [1]-[2], however we don't see the full citation anywhere in the review. Can this be confirmed and updated accordingly?

---

> > ### Comment · Reviewer_X4hj · 2021-11-13
> > **Missing references**
> >
> > Location-Aware Graph Convolutional Networks for Video Question Answering. AAAI 2020
> >
> > Hopper: Multi-hop Transformer for Spatiotemporal Reasoning. ICLR 2021

---

> > > ### Author Response · Authors · 2021-11-13
> > > **Thanks for your update and suggestions!**
> > >
> > > We thank your update and suggestions. We will add the baselines to our paper.

---

> ### Author Response · Authors · 2021-11-15
> **Respose to Reviewer X4hj (2/2)**
>
> **While I understand the intuition about the strategy to extract the time intervals of atomic events (as shown in Figure 3), I do not see how well this strategy can perform in the continuous and more noisy setting. How reliable are the obtained estimates?**
>
> To operate on the noisy setting we rely on learning our convolution to smooth out the noise in the probabilities. The scale of these scores is controlled by learning $\alpha$ post convolution to shift corresponding atomic event noise below $\epsilon$ (Eq 3,4) while keeping high atomic event scores above it. We can extract the predicted time intervals from our trained model and compare to the ground truth intervals within the CATER data. Here we see that the IoU of the predicted intervals over the ground truth object movements was .85, so it captures this interval in the noisy setting well.
>
> **In CATER experiments, is the perception component (i.e. Faster R-CNN) trained jointly to the Neural TLP or are two components trained separately?**
>
> We train these components separately. This was done using bounding box and atomic event annotaitons provided by CATER on key frames as supervision. Using these bounding boxes for object level features, image features, and optical flow to infer the atomic events used for Neural TLP's input. Details are provided in Appendix D.1: Predicted Atomic Events.
>
> **The names parameter and structure learning are a bit confusing in this context. In their classical definition, parameter learning considers learning the parametric relations among variables in a given graph. In Neural TLP, the temporal relations are not given, rather learnt. Why is this considered as parameter learning?**
>
> In our parameter learning step we do **learn** relation parameters $\beta, \gamma$ for our predicate predictions. These are described in more detail in the Predicate Modeling paragraph under Equation 9, where the parameters adapt to start and end time noise and are learned end-to-end. These scale and shift parameters are indeed useful, as we can see freezing them in Table 5 drops Hits@1 from .913 to .681.
>
> During parameter learning we are also learning the convolution kernel $\mathbf{K}$ as well as the scalar $\alpha$. We refer to these learning steps are the paramter learning stage, since these are the necessary paramters to induce the grounded pairwise predicates between atomic events.
>
> The structure learning stage is then learning the structure of the rules, given that the grounded predicates are reliably inferred from earlier. These final rules are composed of conjunction of grounded predicates and are final output of Neural TLP. Nevertheless, we thank you for this opportunity to clarify.
>
> **Can you mention something about the logic expressiveness of the induced rules?**
>
> Currently Neural TLP supports rules containing conjunctions of predicates. It is the first method to induce such rules from *raw* time series data to the best of our knowlege. The limitation about such induction is that we limit the expressiveness we have in our rules. More general conjunctive/disjunctive normal form to support negations and disjunctions would provide more expressive rules. However, more expressive rules typically leads to an even larger search space to consider. Such rules are inevitably useful and provide directions for future work.
>
> ### Video QA
> We are currently working on implementing a video QA encoder using a more structured object-based approach.
>
> ### Closing Comments
> Thank you for giving us an opportunity to clarify our work. Please let us know if you have any further questions.

---

> ### Author Response · Authors · 2021-11-15
> **Respose to Reviewer X4hj (1/2)**
>
> We thank the reviewer for their time and insightful questions. Here we address the specific questions asked by the reviewer.
>
> ### Questions
>
> **It is not clear how the work positions with respect to recent literature in spatio-temporal reasoning and video question answering. For example, the authors should have a look at [1]-[2]**
>
> Recent literature like these focus on leveraging object level structure to better encode object movements over time. For works [1]-[2] the reasoning over these object trajectories comes from Graph Neural Networks and Transformers respectively. From the label optimization perspective, these models work quite well on noisy data. However it is unclear how to lift the underlying decision making, thus rules, from such dense networks. These works do provide a more structured encoder compared to our video baselines, so we will include their mAP results on CATER.
>
> **How scalable is the proposed strategy in terms of number of objects, number of atomic events and compositionality of the target events?**
>
> Our methods scales well with repect to the number of objects, atomic, and composite events. One key contribution is the intermediate inferred atomic event intervals. This allows us to compare the extracted start and end event intervals (Figure 3) across different atomic events as well as across different objects. Using the start and end times we propose parametrized rules to extract the predicates, which are simple to compute as well (Equations 3-9).
>
> The alternative is to retain an embedding or a time series corresponding to each atomic event per object, which can be much higher dimenisonal. This would require a pairwise comparison using a neural based model across all timelines would be more expensive from both memory and computation perspectives.
>
> With respect to the composition of predicates for event rules, we only learn an attention vector to select which rule to choose from our deterministic combinatorial matrix $\mathbf{C}$. Each value in this attention vector simply indicates the score of each combinatorial rule with respect to each composite event label. This means the number of structure parameters increases linearly for each rule in our search space.
>
> In practice for our CATER experiments, we allowed a max of $k=30$ objects and with $|\mathcal{X}|=14$ atomic events we had $30^2 * 14^2=176k$ comparisons between every object and their corresponding atomic events per sample. We were able to handle this inference jointly with structure learning of the composite event rule up to clause length 4 on a modest GPU (2080Ti 11GB). This is quite siginificant for scalability as we optimize over 419M rules per batch step (21,760 rules up to clause length 4 for each 301 composite events per sample and a batch size of 64). We refer to the lemma in Appendix A for the rule space calculation and Appendix B for the rule length optimization details.
>
> **As far as I understand, the proposed strategy considers only the temporal relations among atomic events and objects are considered independent of each other. Is that correct? If so, why is that the case?**
>
> Yes, objects and events are independent of each other. This is the assumption we make because we don't know any prior information about object and event interaction. Therefore our method has to consider all possible predicates between objects and events when constructing rules.
>
> This becomes more interesting as we know more about the objects and events. For example if we know only a subset of objects interact with each other, or certain events are only occuring with a certain set of other events. Then we can reflect this as additional information into the model to determine which comparisons are being made. For example in sports or games we know that the players will only interact with other players, and not the spectators. Therefore we can predefine logic within the model to only make comparisons between objects that are identified as players, since we know this prior knowledge. Good question!

---

### Official Review · Reviewer_54Fe · 2021-11-12

**Correctness:** 3
**Technical Novelty And Significance:** 2
**Empirical Novelty And Significance:** 2
**Recommendation:** 3
**Confidence:** 3

**Main Review:**

Strength

	1.  formulating combinatorial rule space search as predicate selection indicator

Weakness

	1. The proposed model's parameterization depends on the number of events and predicates making it difficult to generalize to unseen events or required retraining.
	2. The writing needs to be improved to clearly discuss the proposed approach.
        3. The experiments baselines are of the authors' own design; it lacks a comparison to the literature baselines  using the same dataset. If there is no such baseline, please discuss the criteria in choosing such baselines.


Details:

	1. Page 1, "causal mechanisms", causality is different from temporal relationship. Please use the terms carefully.
	2. Page 3, it seems to me that M_T is defined over the probabilities of atomic events. The notation as it is used not making it difficult to make sense of this concept. Please consider providing examples to explain M_T.

	3. Page 4, equation (2), it is not usual to feed probabilities to convolution.
		a. Please discuss in section 3 how your framework can handle raw inputs, such as video or audio? Do you need an atomic event predictor or human label to use your proposed system? If so, is it possible to extend your framework to directly have video as input instead of event probability distributions?  Can you do end2end training from raw inputs, such as video or audio? (although you mentioned Faster R-CNN in the experiment section, it is better to discuss the whole pipeline in the methodology).
		b. Have you tried discrete event embeddings to represent the atomic and composite events so as the framework can learn distributional embedding representation of events so as to learn the temporal rules?
	4. Page 4, please explain what you want to achieve with M_A = M_C \otimes M_D. It is unusual to multiple length by conv1D output. Also please define \otimes here. I am guessing it is elementwise multiplication from the context.
	5. Page 4, "M_{D:,:,l}=l. This can be thought as a positional encoding. It is not clear to me why this can be taken as positional encoding?
	6. Page 6, please detail how do you sample top c predicates. Please define what is s in a = softmax(s). It seems to me the dimension of s with \sum_i (c i) can be quite large making it softmax(s) very costly.




**Summary Of The Paper:**

This paper proposed a neural nets based approach for Temporal Logic Programming. A key claimed contribution is on formulating combinatorial rule space search as predicate selection indicator vectors assignment so as to make it differential to enable end2end training from atomic event probabilities as inputs.

**Summary Of The Review:**

This is a good initial attempt to attack the neural temporal logic programming. However, the steps in the proposal --- temporal compression, predicate modeling, composite event prediction, combinatorial inference, and rule reduction --- requires more clarity and refinement to reach a publication state. Overall, the proposal has little element of neural networks. To me it is more on matrix formulation of ILP problem. It is Okay to me but please consider renaming the paper title.

---

> ### Author Response · Authors · 2021-11-15
> **Respose to Reviewer 54Fe (3/3)**
>
> **Page 4, "M_{D:,:,l}=l. This can be thought as a positional encoding. It is not clear to me why this can be taken as positional encoding?**
>
> This can be thought as encoding the time index into the model itself, as discussed above. An analogy is the transformer use case, where the word tokens are presented in a sequential order, but there is no inherent ordering. Therefore they add a positional encoding to differentiate tokens located in different parts of the sentence.
>
> Similarly here we have time series scores for our "tokens" and we want to encode the position information of each score, so we multiply the position into the score itself.
>
> **Page 6, please detail how do you sample top c predicates.**
>
> After we run the parameter learning stage, we have a dense projection matrix $\mathbf{W}$ from the space of all grounded predicates to the space of all composite event labels. Essentialy, this is logistic regression over which grounded predicates are predictive for each composite event. Here it is hard to extract rules from these weights directly, so we first select the top $c$ grounded predicates indicated by weight magnitude. These $c$ predicates are most likely to be the predicates needed to construct the ground truth rule.
>
> Then during structure we search the rule space of combinatorial predicates obtained from these top $c$ predicates from $\mathbf{W}$. This enables us to prune the search space by avoiding the search over irrelevant predicates, indicated by $\mathbf{W}$. We further illustrate this in Figure 4 of our paper.
>
> **Please define what is s in a = softmax(s). It seems to me the dimension of s with \sum_i (c i) can be quite large making it softmax(s) very costly.**
>
> $\mathbf{s}$ is simply the real value vector trained to select the most relevant rule from our combinatorial enumeration of rules $\mathbf{C}$. Here the softmax forces the model to choose a single rule to infer the composite event label. In practice we limit our $c$ to be 100,100,30,25 for rule clause lengths of $n=$ 1, 2, 3, and 4 respectively. This makes softmax for $\sum_{i=1}^n {c \choose i}=21,760$ more tractable to compute in our case.
>
> **However, the steps in the proposal --- temporal compression, predicate modeling, composite event prediction, combinatorial inference, and rule reduction --- requires more clarity and refinement to reach a publication state**
>
> We hope the above responses make this process more clear. We are updating the paper where we can add more clarity.
>
> **Overall, the proposal has little element of neural networks. To me it is more on matrix formulation of ILP problem**
>
> The method indeed has fewer parameters than tranditional deep networks, but this is not necessarily a drawback. Given the space of atomic events we are able leverage strong inductive biases such that these predicates don't have to be learned in a purely neural fashion from data. Rather we leverage known rules of predicate computation (ex. $u_{start} > v_{end} \rightarrow$ after(u, v)) and parameterize them to fit to data. This is critical since many applications don't have a large number of datapoints to use for training, such as videos, or health records in our case (2k samples). We can leverage carefully placed neural components in order to learn smoothing operations like convolution and to infer the labels end-to-end from the computed predicates.
>
> Our current approach uses the *sparse* matrix formulation for ILP. They key contribution is how to populate these sparse matrices by intermediately inferring the predicates in a differentiable manner directly from timeseries data. Learning such ILP problems from *raw* data has not been well explored to the best of our knowledge.
>
> **It is Okay to me but please consider renaming the paper title.**
>
> We will consider this as well. Some title such as: Learning Temporal Rules from Noisy Timeseries Data, would be more direct.
>
> ### Closing Comments
> Thank you for giving us an opportunity to clarify our work. Please let us know if you have any further questions.
>
> [1] Rocktäschel and Riedel. End-to-End Differentiable Proving. NeurIPS'17
>
> [2] De Raedt et al. Inducing Probabilistic Relational Rules from Probabilistic Examples. IJCAI'15
>
> [3] Yang et al. Differentiable Learning of Logical Rules for Knowledge Base Reasoning. NeurIPS'17

---

> ### Author Response · Authors · 2021-11-15
> **Respose to Reviewer 54Fe (2/3)**
>
> **Please discuss in section 3 how your framework can handle raw inputs, such as video or audio? Do you need an atomic event predictor or human label to use your proposed system? If so, is it possible to extend your framework to directly have video as input instead of event probability distributions?  Can you do end2end training from raw inputs, such as video or audio? (although you mentioned Faster R-CNN in the experiment section, it is better to discuss the whole pipeline in the methodology)**
>
> It is theoretically possible, and nice to have such truly end-to-end models. However, there are a few practicality considerations.
>
> The first consideration is the atomic event predictor. Ideally we would leverage pretrained networks to extract atomic events. For example for images this could be pre-trained scene graph detectors to identify objects and relationships. Similarly in video there exists many action recognition models to use, depending on our use case. If this does not exist then we have to generate labels to identify these atomic events. For example, since CATER is synthetic we don't have such atomic event data, so we define label heuristics to approximate the object detection and movements corresponding to atomic events.
>
> The second consideration is the computaiton for such an end-to-end model. For our video task we would be taking raw images, or sequences of images, to predict the atomic events at each timestep. If we have 300 frames as we have in CATER, this would involve storing **all** intermediate image feature and gradients for up to 300 frames. This is intractable to do naively with the downstream considerations of memory required for the rest of Neural TLP. One avenue is engineering to chunk or disribution of the computation to many GPUs, which is possible. Another avenue is downsampling the frames we consider for end-to-end evaluations. Downsampling might not provide optimal results when event intervals are fine grained, as this sampling would lose too much information. This is seen in third row of Table 5 in Appendix D.2, where compressing the event timeline too much starts losing rule inductive performance (Hits@1).
>
> When computing end-to-end, without a pretrained atomic event predictor, it is unlikely that a complex neural predictor will extract the correct relations. Since such an optimization is highly non-linear there are no guarantees that such produced events will be useful downstream for Neural TLP. Therefore for our first attempt, we leveraged FR-CNN to extract the shape and use the frame movements to predict atomic events labels. This was trained independently, and the inferred results were used as our input to Neural TLP. We will add comments about trained atomic event predictors in section 3, per your suggesiton.
>
> **Have you tried discrete event embeddings to represent the atomic and composite events so as the framework can learn distributional embedding representation of events so as to learn the temporal rules?**
>
> Currently we represent the events symbolically, which enables us to compute interpretable intervals, predicates, and rules from there. As we mentioned earlier, it might be possible to use some embedding for entities, but then the following event interval, predicate, and rule representation would be embeddings as well. It is unclear how we would be able extract logic rules from embedded entities, but it is a great avenue for future work.
>
> **Page 4, please explain what you want to achieve with M_A = M_C \otimes M_D. It is unusual to multiple length by conv1D output. Also please define \otimes here. I am guessing it is elementwise multiplication from the context.**
>
> For $\mathbf{M_C}$ we have the post convolution smoothed scores for each atomic action. Say these scores $\in [0, 3]$. Now when computing the interval we need to determine which scores correspond to the start and which correspond to the end. If all the scores belong in the same range, how do we do this? The intution is that if we multiply $\mathbf{M_D}$, which correspond to the time index we can tell between the time points that come earlier and those that come later.
>
> Say we have two scores for an atomic event in $\mathbf{M_C}$, both scores are 1.5. One score occurs at t=10, the other at t=20. Then when multiplying in $\mathbf{M_D}$ the scores become $1.5 * 10=15$ and $1.5 * 20=30$ respectively. From these scores it is much easier to distinguish that the score at 30 comes after 15. Although it seems unusual, this temporal difference is necessary when computing the intervals (Figure 3, Eq 3,4) and predicates (Eq 5-7) dowstream. There we leverage the properties of this time encoder to determine earlier versus later scores. Yes, $\odot$ is elementwise multiplication.

---

> ### Author Response · Authors · 2021-11-15
> **Respose to Reviewer 54Fe (1/3)**
>
> We appreciate the reviewer's comments and suggestions. Here we address these comments directly.
>
> ### Method Questions
>
> **The proposed model's parameterization depends on the number of events and predicates making it difficult to generalize to unseen events or required retraining**
>
> Yes we assume a fixed set of events and predicates. It might be possible to encode some description of each atomic event to represent it in a continuous embedding, where we decode a sequence of embeddings to formulate our rules. However rule extraction from such dense embeddings is non-trivial, but we can take inspiration from methods that do this in the a-temporal setting, such as Neural Theorem Provers [1], for future work.
>
> **The experiments baselines are of the authors' own design; it lacks a comparison to the literature baselines  using the same dataset. If there is no such baseline, please discuss the criteria in choosing such baselines**
>
> To the best of our knowledge there is no baseline that operates over raw timeseries data to generate temporal logic rules. Furthermore temporal rules cannot be learned by directly applying existing proabilistic rule learning frameworks such as ProbFOIL [2] or Nerual LP [3]. This is because in temporal rules we don't assume that the predicates share common variable terms to chain the predicates, and don't contain the relevant start and end entities for each rule (described more in page 2 of the introduction and Appendix A).
>
> This led us to create the Temporal MAP baseline. Here we assume that a preprocessor takes in the time series and processes the predicates corresponding to each sample. This is done in a deterministic fashion for each predicate by thresholding the event probabilities to determing where the event takes place and then using hard rules to predict predicates (ex. $u_{start} > v_{end} \rightarrow$ after(u, v)). Then a baseline temporal rule learning would find the combination of predicates that maximizes the likelihood of each composite event label. This is done through the baseline's maximum a posteriori computation to produce rules for each composite events given the precomputed predicates. For this task this was the most relevant baseline to use for temporal rule induction, and we provide a detailed implementation in Appendix C.2.
> ### Method Details
>
> **Page 1, "causal mechanisms", causality is different from temporal relationship. Please use the terms carefully.**
>
> This makes sense, since we observe temporal correlations, whether they are causal or observational. We make this apparent in our medical results where we emphasize that these are temporal observations. Thanks for the note, and we will update this in our paper draft.
>
> **Page 3, it seems to me that M_T is defined over the probabilities of atomic events. The notation as it is used not making it difficult to make sense of this concept. Please consider providing examples to explain M_T.**
>
> That is correct, in essence we have the probabilities for each atomic event over time $T$, which is the duration of the timeline. There are probabilities for mulitple atomic events $|\mathcal{X}|$, such as different movements: pitch, swing, run, etc. Furthermore we consider multiple objects $k$, such as players, that produce these atomic events. There may be interactions between atomic events across different objects to produce composite events, so we express each objects events explicitly. This provides our full input $\mathbf{M_T} \in [0, 1]^{k \times |\mathcal{X}| \times T}$. We will include this explicitly in the writing.
>
> **Page 4, equation (2), it is not usual to feed probabilities to convolution.**
>
> Since the probabilities are scores for the atomic event, we observe that these scores can be noisy when they have to be inferred, or for example a measurement sensor has noise. When this happens our interval extraction method might incorrectly select the start or end time of the atomic event. Therefore we smooth the probabilities to reduce these errors, which has shown to be effective in first row Table 5 in our ablation Appendix D.2.
>
> Note that these are indeed unnormalized scores post convolution, and we eventually normalize these scores during after our predicate prediction to provide probabilistic predicates.

---

### Author Response · Authors · 2021-11-18
**Summary Response**

We thank all the reviewers for their feedback and we have been working on the suggestions made. We want to briefly summarize the key points made across reviews.

## Writing
We have responded to specific questions reviewers had about our method. Furthermore we spent time updating our manuscript to reflect these suggestions.

A main change we made was in our experimental writing, where we have moved experiments across all baseline and vision models into the main paper results Section 4.1.1. This provides a clear view of our modeling contributions across both rule recovery and classification objectives.

## Rule Recovery Baselines
A major question was if temporal rule learning fits into existing paradigms, and why we chose the baselines we did.

We start with the former by looking at the structure of the rules learned in existing logic programming methods: $f(A, C) := p(A, B) \wedge p'(B, C)$. Here we observe that (1) we know the start and ending entity variables and (2) the predicates between variables for a chain like structure.

Now lets consider a temporal rule, sandwich := before(slice bread, toast) ^ before(wash tomatoes, chop). Here we have a structure of $f := p(A, B) \wedge p'(C, D)$. We see that (1) given a video or large set of actions, we do not know the starting and ending atomic actions and (2) we don't necessarily have a chain between the atomic actions. Therefore we cannot use existing path finding algorithms such as ProbFOIL [2] or Neural LP [1].

We essentially are searching all pairwise combinations between atomic events to construct the rules. Hence we define the Temporal MAP baseline as doing such that, but where predicates are processed deterministically versus stochastically end-to-end in Neural TLP.

## mAP Performance
In CATER we show that LSTMs with atomic event inputs provide the best performance for mAP classification. We discuss how such models may learn spurious correlations in data, which may not align with the underlying mechanisms that led to the label in the first place.

We test this hypothesis by using the CATER rules to synthesize new test timelines containing fewer average composite event labels per sample $\bar{\mathbf{y}}=[10, 40]$. This gives us a different distribution from our original training data where $\bar{\mathbf{y}}=53$. This shows a situation where the data distribution might shift $\bar{\mathbf{y}}=[10, 40]$ and we only have models trained on the source distribution $\bar{\mathbf{y}}=53$. Looking at test mAP on these target distributions, Neural TLP's mAP performance degrades more gracefully than LSTM as the target test data shifts away from the training source data. This shows the power of the underlying inductive baises in our model for generalization.

| Model          | mAP $\bar{\mathbf{y}}=53$ (orig) | $\bar{\mathbf{y}}=40$ | $\bar{\mathbf{y}}=30$ | $\bar{\mathbf{y}}=20$ | $\bar{\mathbf{y}}=10$ |
|----------------|------------------------------------------|-------------------------------|-------------------------------|-------------------------------|-------------------------------|
| Neural TLP     | .69                                      | **.49**                       | **.47**                       | **.45**                       | **.42**                       |
| Attention LSTM | **.75**                                  | .37                           | .34                           | .31                           | .27                           |

### Closing
We hope this summarizes the main points of the reviews so far. We encourage and look forward to more questions and feedback.

[1] Yang et al. Differentiable Learning of Logical Rules for
Knowledge Base Reasoning. NeurIPS'17

[2] De Raedt et al. Inducing Probabilistic Relational Rules from Probabilistic Examples. IJCAI'15

---

### Author Response · Authors · 2021-11-28
**Final Discussion Comments**

Dear reviewers, we have updated the paper and provided responses to your suggestions and questions. Since the discussion period ends soon (Nov 29th), please let us know if there are any further questions we can help with.

---

### Decision · Program_Chairs · 2022-01-20

**Decision:**

Reject

**Comment:**

The paper gives a framework for learning temporal logic rules from noisy unlabeled data. The key novelty is a formulation of combinatorial rule search as an end-to-end differentiable problem. The method is evaluated on a video dataset and a healthcare dataset.

The reviewers liked the high-level ideas behind the paper. However, the conclusion was that the experimental results, while interesting, are still somewhat preliminary (in particular, the baselines are weak).  I agree with this point and am recommending rejection this time around. However, I urge the authors to develop the paper further and submit to the next deadline.